# Deterministic grayscale nanotopography to engineer mobilities in strained MoS$_2$ FETs

Xia Liu [1,10,12] ✉, Berke Erbas [1,12], Ana Conde-Rubio [1,11], Norma Rivano[2,3], Zhenyu Wang [4], Jin Jiang[5], Siiri Bienz[6], Naresh Kumar [6], Thibault Sohier[7], Marcos Penedo [8], Mitali Banerjee [5], Georg Fantner [8], Renato Zenobi [6], Nicola Marzari [2,3,9], Andras Kis [4], Giovanni Boero [1] & Juergen Brugger [1] ✉

Field-effect transistors (FETs) based on two-dimensional materials (2DMs) with atomically thin channels have emerged as a promising platform for beyond-silicon electronics. However, low carrier mobility in 2DM transistors driven by phonon scattering remains a critical challenge. To address this issue, we propose the controlled introduction of localized tensile strain as an effective means to inhibit electron-phonon scattering in 2DM. Strain is achieved by conformally adhering the 2DM via van der Waals forces to a dielectric layer previously nanoengineered with a gray-tone topography. Our results show that monolayer MoS$_2$ FETs under tensile strain achieve an 8-fold increase in on-state current, reaching mobilities of 185 cm²/Vs at room temperature, in good agreement with theoretical calculations. The present work on nanotopographic grayscale surface engineering and the use of high-quality dielectric materials has the potential to find application in the nanofabrication of photonic and nanoelectronic devices.

Semiconducting 2D transition metal dichalcogenides (TMDCs), such as MoS$_2$, are widely investigated for next-generation nanoelectronics and optoelectronics[1,2]. The recent demonstrations of MoS$_2$ transistors with sub-nanometer channels or gate lengths make them encouraging candidates for extending Moore's law[3–5]. However, due to electron-phonon scattering[6,7], a significant limitation for 2DM transistors lies in their charge carrier mobilities. It is thus essential to develop novel strategies for the engineering of the 2DMs' intrinsic properties; these nanofabrication processes need to be compatible with the methods suitable for manufacturing transistors based on these materials.

To enhance the performance of 2DM transistors, numerous approaches have been proposed, each with its own advantages and disadvantages. Channel doping[8–10], contact engineering[2,11], and defect modulation[12,13] have improved the electron mobility of TMDC-based transistors. Yet, intrinsic intervalley scattering critically limits the carrier transport, and thus alternative approaches are needed to engineer the valley structures[14–17]. For silicon electronics, strain engineering is an established approach to modify band structure and carrier mobility, and it is used in production to improve the mobility of silicon-based metal-oxide-semiconductor field-effect transistors (MOSFETs)[18,19]. For

[1]Microsystems Laboratory, École Polytechnique Fédérale de Lausanne (EPFL), 1015 Lausanne, Switzerland. [2]Theory and Simulation of Materials (THEOS), École Polytechnique Fédérale de Lausanne (EPFL), 1015 Lausanne, Switzerland. [3]National Centre for Computational Design and Discovery of Novel Materials (MARVEL), École Polytechnique Fédérale de Lausanne (EPFL), 1015 Lausanne, Switzerland. [4]Laboratory of Nanoscale Electronics and Structures, École Polytechnique Fédérale de Lausanne (EPFL), 1015 Lausanne, Switzerland. [5]Laboratory of Quantum Physics, Topology and Correlations, École Polytechnique Fédérale de Lausanne (EPFL), 1015 Lausanne, Switzerland. [6]Department of Chemistry and Applied Biosciences, ETH Zurich, 8093 Zurich, Switzerland. [7]Laboratoire Charles Coulomb (L2C), Université de Montpellier, CNRS, Montpellier, France. [8]Laboratory for Bio- and Nano- Instrumentation, École Polytechnique Fédérale de Lausanne (EPFL), 1015 Lausanne, Switzerland. [9]Laboratory for Materials Simulations, Paul Scherrer Institute, 5232 Villigen, Switzerland. [10]Present address: School of Integrated Circuits and Electronics, MIIT Key Laboratory for Low-Dimensional Quantum Structure and Devices, Beijing Institute of Technology, Beijing 100081, China. [11]Present address: Institute of Materials Science of Barcelona ICMAB-CSIC, Campus UAB, 08193 Bellaterra, Spain. [12]These authors contributed equally: Xia Liu, Berke Erbas. ✉e-mail: xia.liu@bit.edu.cn; juergen.brugger@epfl.ch

semiconducting 2DMs, strain plays an equally significant role in modifying the band structure and phonon dispersion, thereby influencing scattering processes[20–23]. The use of thermally actuated micromechanical devices[24], substrate heating[25], thermomechanical nanoindentation[20], thin-film stress induced by electric fields[26], bulging devices with pressurized components[21,27], tip indentation to stretch 2DMs via direct contact[28], and the bending and/or stretching of flexible polymer substrates[29–33] have been employed to induce strain and study its physical effects in 2DMs. However, most of these techniques are incompatible with existing silicon-based technologies with high-density integration capabilities. Strain induced by substrate lattice mismatch[34], thermal expansion mismatch[35], integration with thin film stressors[27,36] or underlying thin film stress constrains the choice of substrate. For scalability purpose, strain engineering of atomically thin materials using pre-patterned substrates has been developed[22,37–43]. For instance, crested substrates have been introduced to enhance the performance of optical and/or electrical devices based on 2DMs with induced strain effects. However, the use of pre-structured substrates often results in suspended 2DMs, leading to potentially unstable semiconductor-dielectric interfaces. Therefore, the fabrication of compact nanoscale 2DM transistors with deterministic strain distribution compatible with advanced device architecture has yet to be achieved.

In this work, we demonstrate that monolayer MoS$_2$ FETs subjected to a permanent multiaxial tensile strain on gate oxide patterned with sinusoidal waves achieve significantly higher electron mobilities compared to unstrained devices. The fabrication is based on a sequence of advanced nanofabrication techniques, including thermal scanning probe lithography (t-SPL) for grayscale nanopatterning[44–46]. The t-SPL with a lateral resolution below 10 nm and sub-nanometer vertical depth control enables fabrication of high-resolution grayscale nanopattern with deterministic aspect ratio control[45,47]. The tensile strain in the 2DM is induced through the elongation of the 2DM during the process of contact-transferring a planar 2DM flake using an elastomeric stamp onto a grayscale sinusoidal silicon dioxide (SiO$_2$) dielectric. This sinusoidal topography is previously fabricated by t-SPL and dry etching and can be programmed by adjusting the aspect ratios, also referred to as depth-to-pitch ratios. The depth-to-pitch ratio control capability of t-SPL, which cannot be achieved with such precision through other grayscale nanopatterning techniques such as electron beam lithography[48] and interference lithography[49], provides deterministic control of strain induced in 2DMs. Varying the depth-to-pitch ratios of nanotopographies allows for the introduction of areas with strain gradients within a single 2DM flake by adjusting the amplitude and spatial frequency of the sinusoidal waves. Compared to other approaches such as nanopillar arrays and rippled or crested substrates, grayscale nanotopographies also offer improved conformal attachment of 2DMs by reducing wrinkles and suspended parts. This results in improved dielectric-semiconductor interfaces and a mechanically more stable environment compatible with subsequent fabrication processes. In contrast to the 2DMs strained by sharp crested patterns, where strain is non-uniform and is very high at peaks and very low on flat parts, grayscale nanopatterns offer a more homogeneous distribution of strain while still keeping the strain localized within the pattern area. We systematically study how the tensile strain affects the electrical performance of 2DM transistors. We find noticeable enhancements both in electron mobility and in on-state current of the strained 2DM transistors, up to 8 times compared to the unstrained ones. First-principles calculations of electron-phonon scattering, which consider doping and valley profile as parameters, are used to estimate theoretically the effect of strain on electronic transport. The theoretical findings are in good agreement with the experimental results and predict the enhancement of electron mobilities in the presence of strain obtained by surface topography engineering of the gate dielectric.

## Results and discussion

### Design and fabrication of strained 2D FETs

When exposed to tensile strain, a MoS$_2$ monolayer undergoes an expansion in lattice parameter and in out-of-plane atomic displacements (see Fig. 1a). This tensile strain translates into an increased energy separation between the K- and Q-valleys, as shown in Fig. 1b, blocking some of the available electron-phonon scattering channels and resulting in a reduction of electron-phonon scattering[23] and consequently in increased mobility[14,15]. Fig. 1c shows the configuration of the strained monolayer MoS$_2$ FET, where the MoS$_2$ layer adheres conformally via van der Waals forces to a nanopatterned SiO$_2$ gate dielectric. The surface of the gate dielectric is patterned with sinusoidal wave topographies to introduce tensile strain in the monolayer flake of MoS$_2$ placed on top (Supplementary Fig. 1). During the transfer to the topographically shaped gate dielectric, the MoS$_2$ flake deforms along multiaxial in-plane directions of the sinusoidal waves, leading to a tensile stress for the 2DM lattice.

To demonstrate the effect of strain on the carrier transport, we fabricated tensile strained FETs made from monolayer MoS$_2$ flakes, following the process flow detailed in Fig. 1d. First, a thermally sensitive resist, polyphthalaldehyde (PPA), is spin-coated on the SiO$_2$/Si chip. Then, biaxial sinusoidal waves with a diagonal pitch of $300\sqrt{2}$ nm and varying peak-to-peak diagonal amplitudes are patterned on PPA by t-SPL (Fig. 2a and Supplementary Fig. 2). Since PPA is not a reliable gate dielectric, we transferred the pattern by dry etching into the underlying SiO$_2$ layer (Fig. 2a and Supplementary Fig. 3). This process allows introducing a variability of strain gradients into the 2DM by adjusting the amplitude of the nanotopography (see Fig. 2b). To limit surface roughness that could cause nano-sized cracks in the transferred 2D flakes, a low etch selectivity between SiO$_2$ and PPA of 0.7 is developed to transfer the pattern into SiO$_2$ dielectric, resulting in a surface roughness of ~ 1.2 nm$_{rms}$ (Fig. 2c). The 2DM layer is then transfer printed using a polymer support, whereby the 2DM deforms and adheres to the wavy SiO$_2$ surface by van der Waals forces. After removal of the transfer polymer, the 2DM remains in conformal contact with the wavy surface. To minimize the risk of the 2DM sliding on the SiO$_2$ surface during the transfer step that would result in a subsequent release of deliberately induced intrinsic stress, we designed the wavy area of the substrate to be smaller than the 2DM flake (Fig. 2d), effectively mitigating sliding.

To structure the MoS$_2$ flakes into a transistor configuration, we lithographically patterned the 2DM in order to obtain isolated FET channels. Source (S) and Drain (D) electrodes are subsequently created by lift-off metallization of Ti/Pt. To protect the 2DM layer during the fabricating process, it is covered by a PMMA resist layer until the final lift-off step (Supplementary Figs. 8–10). The SEM image in Fig. 2e shows the nanopatterned MoS$_2$ channel and the S/D electrodes.

We fabricated 75 strained FETs and 90 flat FETs on a single chip (1 cm × 1 cm), as shown in Fig. 2f, with the fabrication process detailed in Supplementary Fig. 9. To be able to validate our hypotheses it is of utmost importance that we can assert that the 2DM follows the nanotopographic contours conformally. We thus systematically investigated the 2DM/dielectric interface using various characterization techniques. First, SEM and cross-sectional TEM along with energy dispersive X-ray (EDX) elemental maps were performed (Fig. 2e, g and Supplementary Fig. 11). This was followed by quantitative nanomechanical AFM mapping in various imaging modes (Fig. 2h–j). The adhesion map consistently reveals a sinusoidal pattern, whether 2DM is present or not (Fig. 2i). Notably, the presence of 2DM is scarcely discernible in both the topography and deformation maps, which are characterized by relatively high tip forces of 10 nN for monolayer flakes, indicating a strong adhesion between the 2DM and the dielectric surface (Figs. 2h and 2j). However, a few wrinkles with tens of nanometers width are also shown in Fig. 2j. All characterization steps undertaken confirm the conformal contact of the 2DM layer with the underlying wavy SiO$_2$ dielectric.

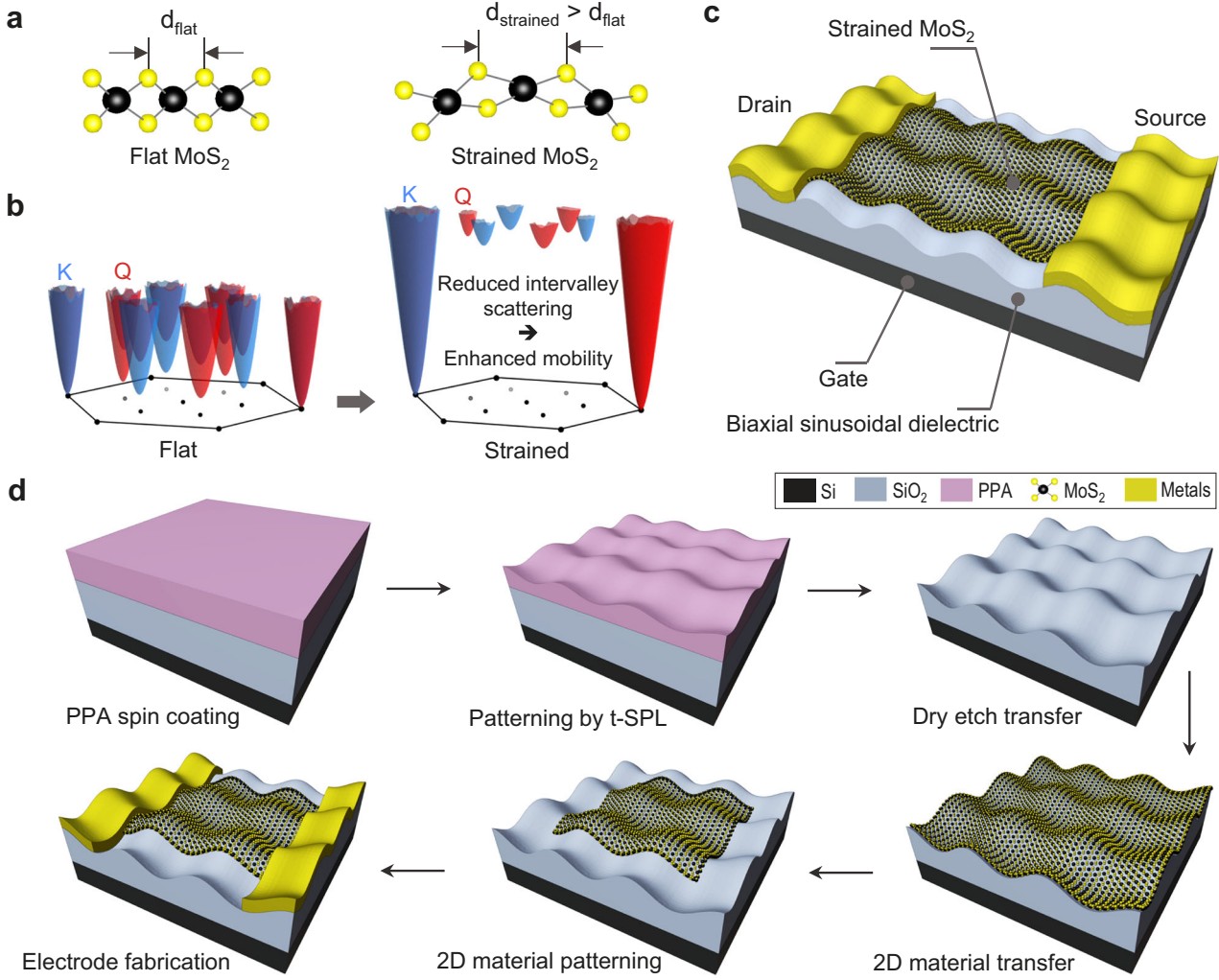

**Fig. 1 | Concept drawing of a strain-engineered 2D transistor made by grayscale nanopatterning of the gate dielectric. a** Atomic structures of the flat MoS$_2$ (without strain) and the MoS$_2$ with tensile strain illustrating the variation in lattice parameters. **b** Density-functional theory (DFT) calculations of the electron spin-valley landscape of MoS$_2$, showing the difference in the K- and Q-valley profiles arising under tensile strain and resulting in reduced intervalley scattering. **c** 3D illustration of a strained monolayer MoS$_2$ FET with a sinusoidally shaped gate dielectric, inducing tensile strain in the transistor's two-dimensional material (2DM) channel. **d** Device fabrication steps: polyphthalaldehyde (PPA) spin coating, thermal scanning probe lithography (t-SPL) patterning, dry etch transfer, 2D material transfer, 2D material patterning by electron beam lithography (EBL) and dry etching, and electrode fabrication by EBL and metal evaporation.

## Strain characteristics of strained 2DM transistors

Introducing tensile strain in 2DMs without causing substantial material damage (e.g., folds or cracks) is one of the key assets of our approach. As illustrated in Supplementary Fig. 14, the monolayer MoS$_2$ flake is picked up with a PDMS or PC film and then transferred onto the target pre-patterned gate dielectric. Micro-Raman spectroscopy of the nanopatterned MoS$_2$ shows a clear redshift of the $E^1_{2g}$ peak as a result of the induced tensile strain (Fig. 3a). The Raman measurements are performed at three different positions along the channel, resulting in consistent redshifts. The splitting of the Raman peak $E^1_{2g}$ indicates significant strain in the nanopatterned MoS$_2$ transistors. However, no shift in the Raman peak $A_{1g}$ is observed. An increase in the $A_{1g}$ FWHM of MoS$_2$ also indicates the effect of strain[31] as compared in Supplementary Table 1. To determine the reproducibility of our method, we examined an array of transistors fabricated on the same flake and found similar redshifts (Fig. 3b and Supplementary Fig. 15). The strain in the transistor is derived from the biaxial sinusoidal wave pattern engineered in the substrate prior to the 2DM transfer printing. As expected, increasing the SiO$_2$ pattern amplitudes results in a proportional increase in the induced tensile strain on the MoS$_2$ layer, corresponding to the surface area increase. In our design, we implemented

wavy nanopatterns with peak-to-peak amplitudes up to 65 nm and a diagonal pitch of $300\sqrt{2}$ nm, resulting in up to 5.6% increase in surface area per unit design area. With this configuration, we anticipated a theoretical strain of up to 2.8%, calculated using the strain formula based on elongation (Supplementary Fig. 16). In our fabricated sample, we measured a shift of the $E^1_{2g}$ peak by $-4.4$ cm$^{-1}$, corresponding to a strain of 1% based on the widely reported and theoretically predicted Raman peak shift of $-4.5$ cm$^{-1}$/% strain for $E^1_{2g}$ phonon[22,30]. However, other works have reported a range of values for the peak redshifts ranging from 2.1 cm$^{-1}$/% strain[50] to 5.2 cm$^{-1}$/% strain[37] for $E^1_{2g}$ phonons. These minimum and maximum values lead to an estimated strain range from 0.85% to up to 2.10% when our measured Raman shifts are considered. While it is possible to derive intermediate values from existing literature[20,22,30], we opted to utilize the extensively documented and theoretically anticipated Raman peak redshift of 4.5 cm$^{-1}$/% strain for the $E^1_{2g}$ phonon as our benchmark. The discrepancy between the theoretically calculated strain, which is related to surface area increase through sinusoidal nanopatterning, and measured strain might arise from several factors, including flake sliding and strain relaxation during transfer, as well as imperfect attachment of the 2DM on a sinusoidal surface with a high spatial frequency

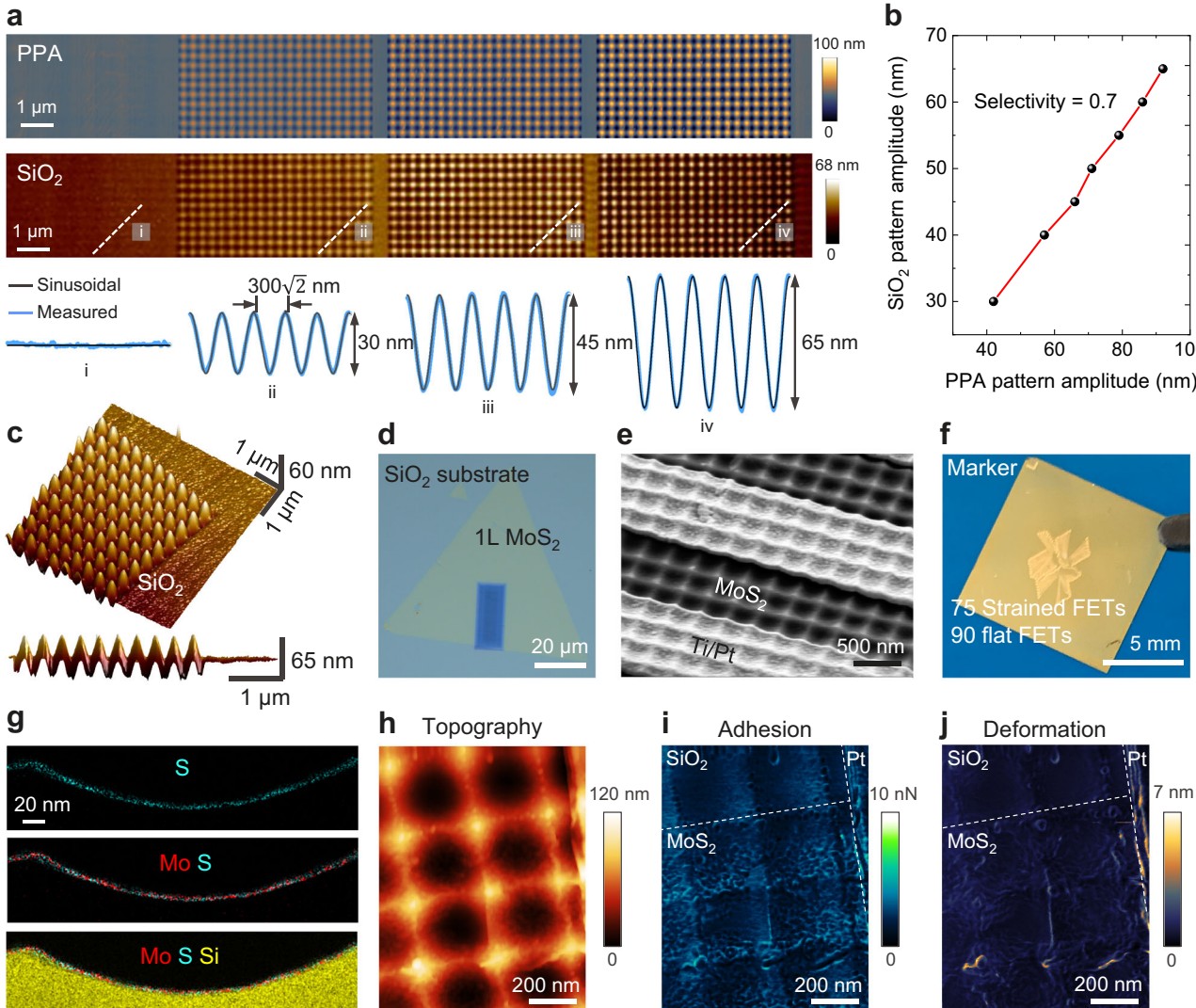

**Fig. 2 | Fabrication and characterization of strained monolayer MoS₂ transistors. a** Biaxial sinusoidal pattern of PPA resist with varying depth profiles and the corresponding patterned SiO₂ substrate fabricated by dry etching-based pattern transfer from PPA nanopatterns, with its measured depth profiles in the selected regions showing periodic sinusoidal waves. **b** Graph showing the peak-to-peak diagonal amplitude of the sinusoidal nanopattern after pattern transfer into SiO₂ by plasma etching with a SiO₂ to PPA etch selectivity of 0.7 favoring smooth surfaces. **c** Atomic force microscope (AFM) image showing the 3D topography of the biaxial sinusoidal SiO₂ substrate having a surface roughness of ~1.2 nm_rms. The side view of the topography shows the regular symmetry of the sinusoidal wave periods. **d** Optical microscopy image of the metal-organic chemical vapor deposition (MOCVD) grown MoS₂ flake transferred on the pre-patterned substrate. **e** Scanning electron microscope (SEM) image of a representative device showing the MoS₂ channel and S/D electrodes both with the biaxial sinusoidal pattern surface. Image tilt angle is 45°. **f** Optical image of a fabricated chip (1 cm × 1 cm) consisting of 75 strained FETs and 90 flat FETs. **g** Element mappings of the transistor cross-section between 2DM and dielectric showing continuous wavy monolayer MoS₂ layer that has intimate contact with the corrugated SiO₂ substrate. **h** AFM topography (height) showing no difference in pattern amplitudes between the flake and SiO₂ substrate. **i** AFM image showing the adhesion map of the strained MoS₂ transistor indicating the position of the 2D flake. **j** AFM deformation image indicating conformal contact between the 2D flake and SiO₂ substrate with exception of a few wrinkles with ≤ 6 nm deformations. Dashed lines represent the borders of materials. Peak Force setpoint is 10 nN for AFM characterization.

pitch. During the elongation of the 2D flake, the sliding between the 2D layer and the sinusoidal patterned substrate is inevitable due to the weak van del Waals force[51], which is one of the critical challenges in strain engineering of 2DMs. The transfer of 2D flakes involves a few temperature-related steps (see Supplementary Figs. 4, 7) that cause strain relaxation of the 2D flakes. The 1% strain achieved remains relatively high, which is sufficient to approach the upper mobility limits in MoS₂ according to first-principles calculations.

We used micro-Raman spectroscopy to visualize the strain distribution as a map, confirming that strain is present throughout the entire nanopatterned 2DM without significant local variations (Supplementary Fig. 17). To visualize the strain distribution below the optical diffraction limit, we performed high resolution AFM-based

tip-enhanced Raman spectroscopy (AFM-TERS)[52]. Fig. 3d shows the AFM topography of a strained 2D FET with the S/D electrodes and the patterned 2D channel. Hyperspectral line TERS mapping was performed across the nanopatterned channel in the rectangular area marked in Fig. 3d, with a step size of 50 nm. The AFM topography data captured simultaneously with the TERS data is presented in Fig. 3e. Figure 3f shows the average TERS spectra, computed from 4 pixels covering an area of 100 × 100 nm², of the patterned MoS₂ on the bottom, the top and the slope of the sinusoidal wave pattern, in comparison to the spectrum of the flat unstrained MoS₂. The TERS spectra of the strained MoS₂ show a splitting of the $E^{1}_{2g}$ band into $E^{1-}_{2g}$ and $E^{1+}_{2g}$ bands, consistent with the results obtained from micro-Raman measurements. Notably, the $E^{1-}_{2g}$ and $E^{1+}_{2g}$ bands are found to

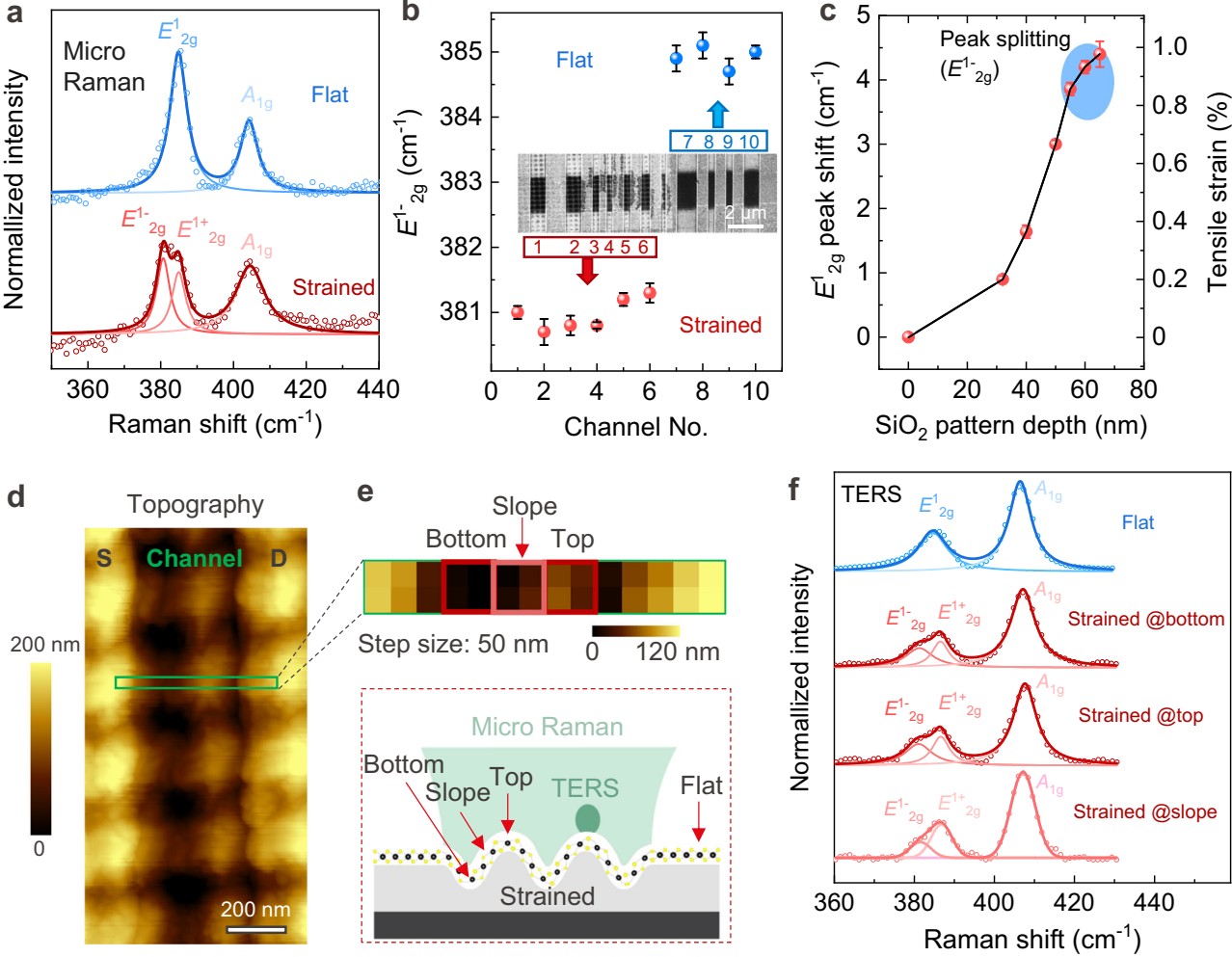

**Fig. 3 | Strain characterization of strained monolayer MoS₂ transistors. a** Raman spectra of the strained and unstrained MoS₂ channels. The circles represent the raw data and the solid lines represent the Lorentzian fit. **b** $E^1_{2g}$ Raman peak position variation in different channels of the same flake. The inset is the SEM image of the strained and unstrained transistors made from the same monolayer MoS₂ flake. **c** Variation of $E^1_{2g}$ Raman peak position as a function of the pattern amplitude for 0, 30, 40, 50, 55, 60, and 65 nm. The blue shaded area represents data from peak splitting measurements. The error bars in panels **b** and **c** represent the maximum and minimum values with the representative points corresponding to the average values. **d** AFM topography showing the strained transistor composed of sinusoidally patterned structures supporting the monolayer MoS₂ sheet as channel and the metal electrodes as source and drain. The region of tip-enhanced Raman spectroscopy (TERS) mapping is highlighted with a green rectangle. **e** Topography image measured during hyperspectral TERS mapping with a step size of 50 nm. The position of the sinusoidal wave structure on the bottom, the top and the slope is marked with red squares and illustrated in the scheme below where the resolutions of micro-Raman spectroscopy and TERS are compared. **f** Comparison of the average TERS spectra of the flat region and the strained regions on the bottom, the top and the slope. A splitting of the $E^1_{2g}$ peaks is observed in the strained region, confirming the presence of strain in the MoS₂ sheet.

be separated by about 5 cm⁻¹ both on the bottom, the top and the slope of the patterned MoS₂, as shown in Fig. 3f. This indicates that both techniques, micro-Raman and AFM-TERS, measure a similar level of strain in the MoS₂ layer, and this consistent strain level is observed on the bottom, the top and the slope of the sinusoidal nanopatterns. Additional TERS measurements from other patterned structures are shown in Supplementary Fig. 18.

### Electrical characteristics of strained transistors

The electrical performance of the strained and unstrained transistors is experimentally measured using a semiconductor analyzer, as shown in Fig. 4a. Figure 4b compares the transfer curves (drain-to-source current, $I_D$ versus gate-to-source voltage, $V_{GS}$) of the strained and flat transistors patterned within a single monolayer MoS₂ flake. The gate leakage currents in all the measured transistors are in the order of a few pA, i.e., much smaller than the drain currents in the 'on' region. The output characteristics ($I_D$ versus drain-to-source voltage, $V_{DS}$) of both strained and flat transistors show a nearly linear regime of $V_{DS}$ in the

range of −0.5 to 0.5 V at room temperature ($T = 300$ K) (Fig. 4c) and at lower temperatures ($T$ ranging from 80 to 260 K) (Supplementary Figs. 21, 22), indicating a nearly ohmic behavior with low Schottky barriers in 2DM-metal configuration (Supplementary Figs. 23–25). Approximate current saturation of both devices is observed at $V_{DS} = 3$ V. The forward and reverse transfer curves of the strained transistor are plotted together on a logarithmic scale in Fig. 4d. Importantly, the devices show a hysteresis-free transfer curve across a wide range of $V_{GS}$ and demonstrate good reproducibility and stability (Supplementary Figs. 28, 29), thanks to the high-quality interface contact between 2DM and dielectric. This provides an additional and convincing demonstration that the flake transfer strategy and the design of the gate dielectric topography are compatible with standard transistor fabrication processes, which in turn preserves the potential for integration of strained 2DM FETs in advanced electronic devices.

The threshold voltage ($V_T$) of the strained transistors is lower than that of the flat ones (Fig. 4e and Supplementary Fig. 20),

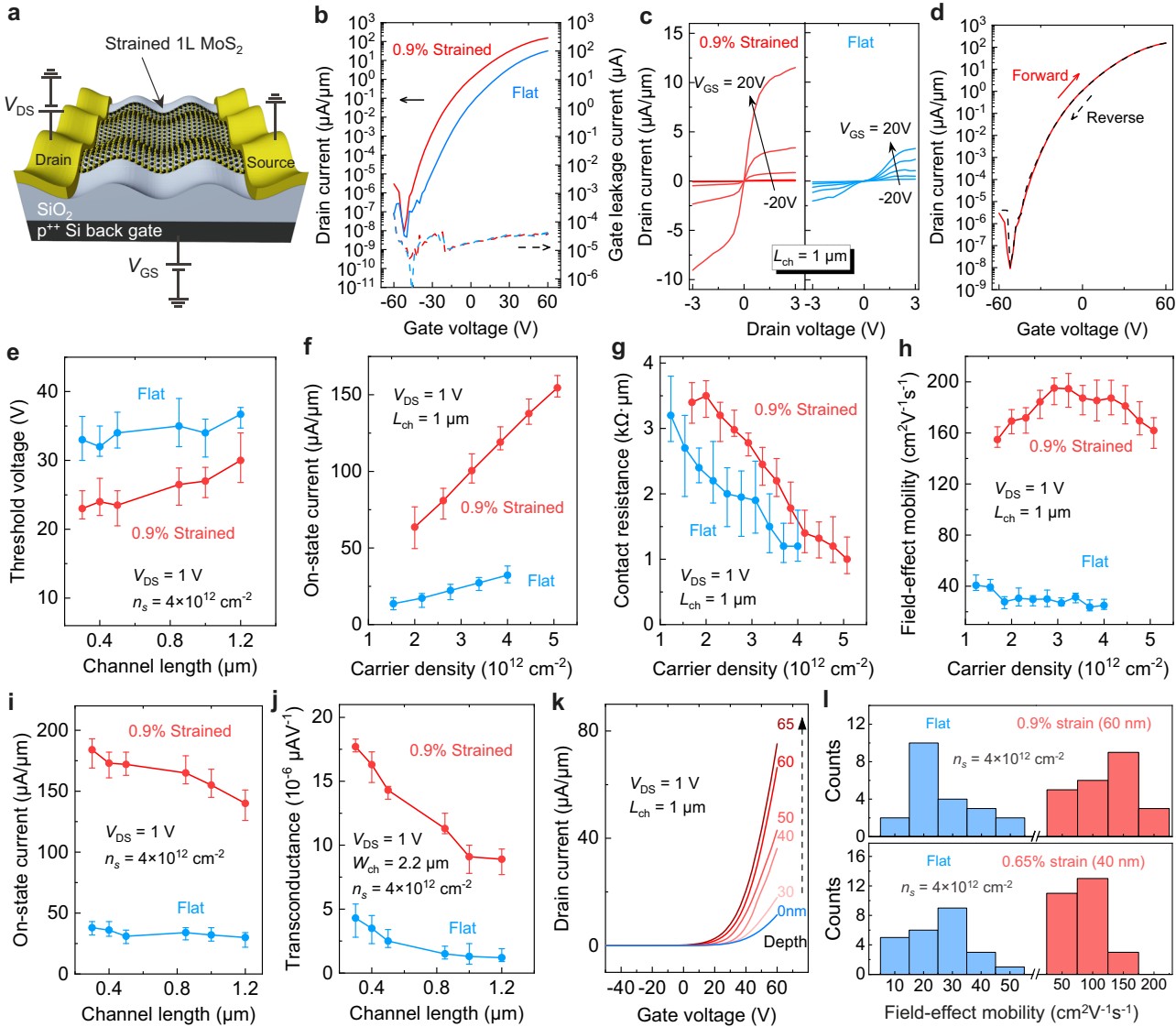

**Fig. 4 | Electrical characteristics of the strained transistors. a** Schematic cross-section and electrical connections of a back-gate strained metal-organic chemical vapor deposition (MOCVD) grown monolayer MoS2 FET. The structure comprises the heavily p-type doped Si substrate as global back-gate, and a dry thermally grown $SiO_2$ as back-gate dielectric. **b** Room-temperature transfer curve of the strained FET measured at drain-to-source bias voltage ($V_{DS}$), $V_{DS} = 1\,V$. The gate leakage current ($I_{GS}$) versus gate-source voltage ($V_{GS}$) is plotted as well. **c** $I_D$ as a function of $V_{DS}$ at varying $V_{GS}$ for strained (left) and unstrained (right) transistors fabricated on the same MoS2 flake with the same channel length and width. **d** Forward (red solid curve) and reverse (black dash curve) transfer curves on logarithmic scales. The curves are hysteresis-free in the entire voltage range of the device after thermal annealing. **e** Threshold voltage ($V_T$) varying with channel length at $V_{DS} = 1\,V$ and carrier density at $n_s = 4 \times 10^{12}\,cm^{-2}$. **f–h** Transistor metrics (**f**: on-state current, $I_{on}$, **g**: contact resistance and **h**: field-effect mobility) of strained ones and flat ones are compared with the varying carrier density at $V_{DS} = 1\,V$. **i**, **j** $I_{on}$ and transconductance ($g_m$) of both transistors are compared with varying channel lengths. The error bars in panels **e–j** represent the maximum and minimum values with the representative points corresponding to the average values for 120 transistors in panels **e**, **i**, and **j**, and for 30 transistors in panels **f–h**. **k** Typical $I_D–V_{GS}$ of strained FETs (red) and flat FET (blue) on $SiO_2$ dielectrics with various pattern amplitudes. **l** Statistical distribution of electron mobility of the strained FETs on the substrate with different strains (or pattern amplitude).

suggesting that the electron density increases with the decreasing bandgap under tensile strain. To account for shifts in $V_T$, we compared transistor metrics, i.e., on-state current ($I_{on}$), contact resistance ($R_c$) and field-effect mobility (Fig. 4f–h) at the same carrier density calculated using $n_s = C_{ox}(V_{GS} - V_T)\,/\,q$ where $C_{ox}$ is the gate dielectric capacitance and $q$ is the elementary charge. The $I_{on}$ of strained and flat transistors linearly increases with the carrier density. The $R_c$ of both strained and unstrained FETs is less than $2\,k\Omega \cdot \mu m$ at $n_s \geq 4 \times 10^{12}\,cm^{-2}$, indicating that the devices are channel-dominated (Fig. 4g and Supplementary Fig. 26). For a fair comparison, the carrier-density-dependent field-effect mobility of both transistors was compared at the same carrier density (Fig. 4h). We also compared the transconductance of both transistors for different channel

lengths (Fig. 4j). Key parameters of the strained and unstrained FETs are shown in Supplementary Tables 2, 3 [53].

With the electrical characterization of both strained and unstrained transistors performed, we next analyze the effect of strain on the field-effect electron mobility and on-state current. By increasing the pattern amplitude from 30 nm to 65 nm and thereby intensifying the strain effect, a noticeable enhancement in the drain current density (Fig. 4k) is observed. To avoid flake-to-flake variabilities, we only compared devices (strained and unstrained) fabricated within same single flakes. To show the reproducibility of the electron mobility enhancement of MoS₂ transistors under tensile strain, we fabricated and measured around 400 transistors made from 8 flakes. The statistical distribution of electron mobilities, measured in flat (unstrained),

0.65%-strain, and 0.90%-strain transistors, is summarized in Fig. 4l. The results are in the range of 8 to 60 cm²/Vs, 50 to 150 cm²/Vs, and 50 to 180 cm²/Vs at 300 K, respectively, with the strained transistors exhibiting a clear increase in electron mobility compared to the flat ones. The mobilities of the flat 2DM transistors do not exceed 60 cm²/Vs, consistent with data in the literature for MoS$_2$ FETs without channel encapsulation[11,54,55].

Analytical calculations and simulations of the electrical field distribution and capacitance in the strained transistors show no significant difference in the gate capacitance, indicating that it exerts no discernible impact on the strained transistors. While flat transistors exhibit a gate capacitance of $2.47 \times 10^{-8}$ F/cm², that of strained transistors with a 60 nm peak-to-peak depth is $2.56 \times 10^{-8}$ F/cm² (Supplementary Figs. 30, 31). The variation in gate dielectric thickness leads to differing doping concentrations in the FET channel, impacting mobilities. However, our comparison is based on flat dielectrics with a 140 nm gate dielectric thickness and sinusoidal dielectrics with a mean thickness of 140 nm, both having the same surface quality. This results in a doping concentration change of only 3.5%, which does

not introduce significant alterations, according to theoretical and experimental studies[56]. Consequently, our findings let us conclude that the mobility enhancement primarily arises from the drastic reduction in intervalley scattering due to the induced strain.

Quantitative analysis of the electron mobility enhancement under various strains is summarized in Fig. 5, with Fig. 5a showing a linear increase in mobility enhancement with strain, followed by a gradual saturation. A similar tendency is observed for both types of flakes, whether they are metal-organic chemical vapor deposition (MOCVD) grown or mechanically exfoliated. Figure 5b shows a comparison of the electron mobility enhancement of the 0.90% strained transistors under different electrostatically induced carrier densities. Linear growth is found as a function of carrier density for both types of flakes. To further investigate the origin of the main scattering mechanisms in the strained transistor, the mobility as a function of temperature is shown in Fig. 5c. In flat MoS$_2$ transistors, the mobility decreases as expected slightly with increasing temperature with a power exponent $\gamma$ of 1.6 at higher temperatures ($T > 200$ K), which is typical for phonon-limited transport[57,58]. However, in the case of strained MoS$_2$ transistors,

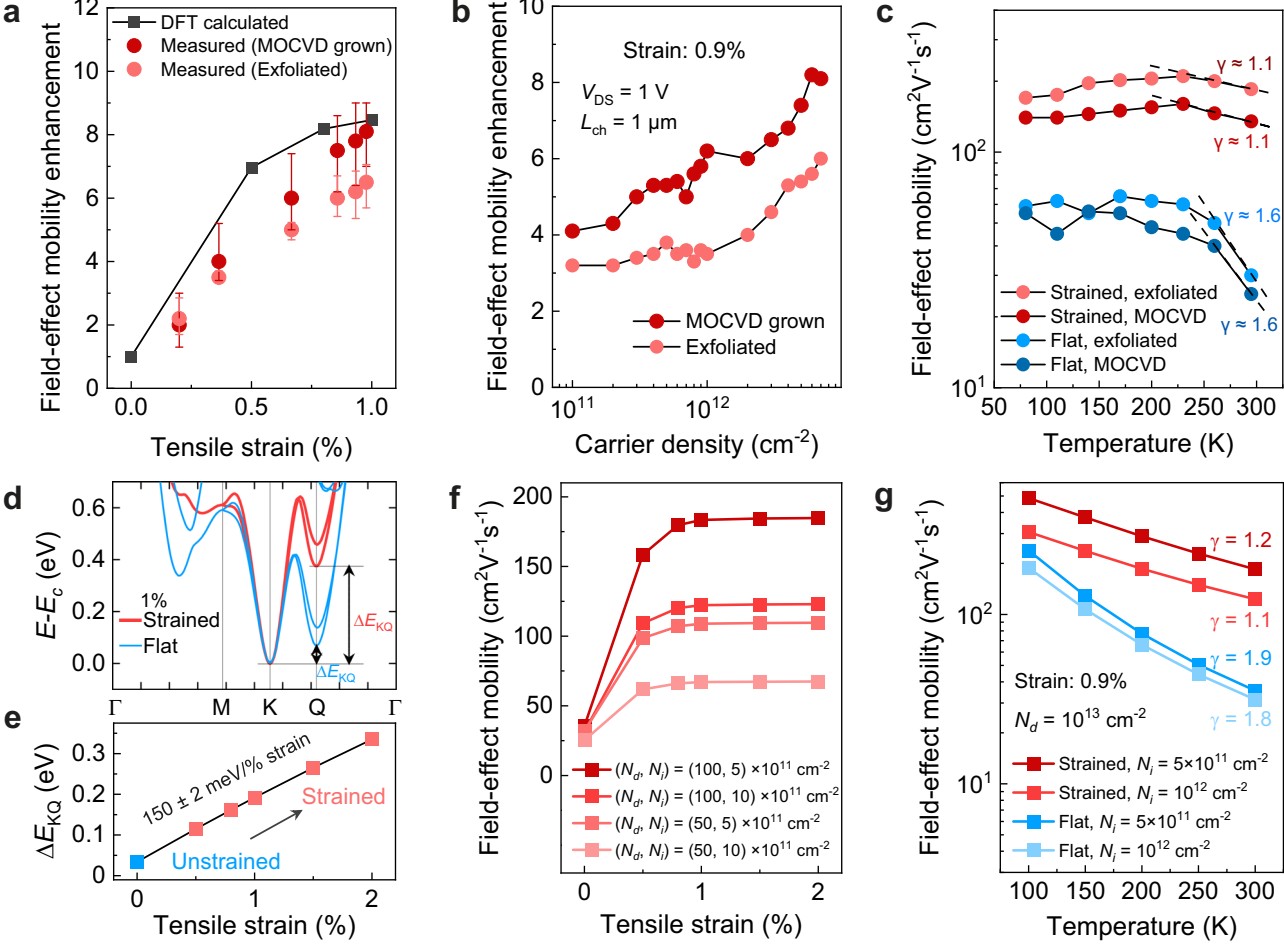

**Fig. 5 | Effect of strain on mobility: experiments and DFT calculation. a** Mobility enhancement as a function of tensile strain for the strained exfoliated flake and strained MOCVD grown flake, comparing with the DFT-calculated value at doping concentrations of $5 \times 10^{12}$ cm⁻², excluding impurity effects to illustrate the enhancement limit for phonon-limited mobility. The error bars represent the maximum and minimum values with the representative points corresponding to the average values. **b** Mobility enhancement as a function of carrier density for the strained exfoliated flake and strained MOCVD grown flake. **c** Electron mobility as a function of temperature fitted with the power law $\mu \approx T^{\gamma}$, where electron-phonon scattering transport is suppressed in strained FETs compared to flat FETs. **d** A zoom

on the band structure of the K and Q valleys for flat (blue) and strained (red) MoS$_2$ with an electron doping of $N_d = 5 \times 10^{12}$ cm⁻². In each case, the two band structures are aligned with respect to the bottom of the conduction band ($E_c$). **e** Variation of valley energy separation, $\Delta E_{KQ}$ as a function of the applied strain. See Supplementary Fig. 19 for the experimental results on the strain induced bandgap modulation obtained through photoluminescence (PL) measurements. **f** Analysis of DFT mobility variation under strain, considering both phonon and impurity scattering, across varying impurity and doping concentrations. **g** DFT mobility (with phonon and impurity scattering) as a function of temperature for the highest experimental doping, $N_d = 10^{13}$ cm⁻², in flat (blue) and strained (red) systems.

the temperature exponent changes its value to $\gamma \approx 1.1$, which indicates a substantial reduction of electron-phonon scattering in strained $MoS_2$ at room temperature.

## Comparison of experimental results with calculations

We support the experimentally measured improvements in electron mobility under strain with extensive first-principles calculations of electron-phonon coupling and mobilities by applying uniform tensile strain across the material. To this aim, we note that the temperature dependence of the mobility (Fig. 5c) reveals distinct contributions to the total enhancement: extrinsic disorder-related mechanisms dominate at low temperatures ($< \sim 150$ K), while intrinsic phonon scattering prevails at higher temperatures. We first study the intrinsic phonon-limited mobility using density-functional perturbation theory (DFPT)[59,60] and the Boltzmann transport equation (BTE)[61,62], establishing an ideal limit for the potential enhancement through strain. We then include a model for disorder[16] and show a decrease in the enhancement towards experimental values (Fig. 5a). Details on the model and calculations are reported in Methods and Supplementary Information.

Considering phonon-limited mobility, we note its strong dependence on the occupation of $MoS_2$'s Q valley[14]. This is attributed to increased electron-phonon scattering through activated intervalley scattering and enhanced intravalley coupling[14,15,63]. The Q valley occupation depends on doping and the relative energy positions of K and Q valleys, $\Delta E_{KQ}$. Straining the 2DM results in a reduction of the layer thickness under tensile strain[64–66], which increases the energy separation $\Delta E_{KQ}$, as sketched in Fig. 1c, depleting the Q valley of electrons and enhancing the phonon-limited mobility.

We also establish that effective masses, phonons, and electron-phonon interactions exhibit negligible variations under strain (Supplementary Information Section 8). Thus, the main changes in phonon-limited mobility arise from strain-induced shifts in the valley positions. We compute $\Delta E_{KQ}$ as a function of strain (Fig. 5e) and use a previously reported ab-initio model[14] to determine the corresponding changes in mobility at doping concentrations of $5 \times 10^{12}$ and $1 \times 10^{13}$ cm$^{-2}$ at room temperature. The energy separation variation rate is around 150 meV/% strain, close to the experimental value obtained from the photoluminescence measurement (Supplementary Fig. 19). The model was precisely designed to describe the changes in electron-phonon scattering as a function of doping and $\Delta E_{KQ}$. Our results show a strain-induced mobility enhancement up to 12.7 and 8.5, respectively for the highest and the lowest electron doping, and mobilities larger than that found in experiments for similar doping (Fig. 5a), establishing an ideal upper limit. Notably, the increase saturates rapidly at modest strains ($\sim 0.8$–1%) as the Q valley is fully emptied. The saturation point depends on the initial $\Delta E_{KQ}$ at 0%; a larger separation requires less strain for maximum enhancement.

Disorder is mainly induced by charged impurities that involve local distortions in the scattering potential, predominantly affecting low temperatures. For more realistic comparisons with experiments, we incorporate electron scattering from charged defects into our calculations[16], adopting typical densities $N_i$ of $5 \times 10^{11}$ cm$^{-2}$ and $1 \times 10^{12}$ cm$^{-2}$, following the literature[67,68]. These are meant to provide an effective model for unknown sources of disorder, notably excluding strain-induced modifications in impurity scattering. In Fig. 5f, we compare strain-dependent mobilities for two different values of doping and impurity concentrations. At constant doping, including impurities significantly reduces mobilities and brings them closer to experiments. Although the saturation point at $\sim 0.8$–1% strain remains, the overall enhancement (5.2–3.9) slightly underestimates experimental data (Fig. 5a, f). This could be due to the simulation's fixed doping and impurity concentrations, whereas experimental fluctuations in these levels under varying strain would lead to deviations from theoretical predictions. At saturation, with an empty

Q valley, mobility can be further boosted by increasing doping or reducing impurities. In flat systems, doping and defects play a smaller role, as the Q valleys are filled and intervalley scattering dominates, explaining why the curves join towards zero strain in Fig. 5f. While some uncertainty remains about disorder, we further characterized the phonon contribution via the power exponents of the computed temperature-dependent mobilities in Fig. 5g. These findings agree with our experimental results shown in Fig. 5c, particularly within the same high-temperature range. These calculations confirm that the increased energy separation due to strain in the 2DM effectively reduces electron-phonon scattering, consequently enhancing electron mobility. Importantly, defects within the $MoS_2$ layer itself and at the underlying gate dielectric interface, which cause extrinsic scattering, are also predicted to play a role in the variations of mobility under strain.

We demonstrated that tensile strain effectively modulates the band structure of $MoS_2$, splitting the K and Q valleys and greatly reducing intervalley phonon scattering in the latter. This significantly enhances electron mobility and provides a powerful approach to extending the performance limits of 2DM transistors. To achieve this, we developed a nanoengineering process for fabricating permanently strained 2DM transistors at the nanoscale. We used grayscale nanolithography based on thermal scanning probe lithography and dry etching to create 2D sinusoidal waves on $SiO_2$ substrates, providing controlled tensile strain in $MoS_2$ transistors at predetermined locations by design. Precisely patterned surface topography at the single-digit nanometer scale enables deterministic changes in the tensile strain induced in 2D materials, thereby locally altering their electrical and optical properties while offering a seamless device integration option. As a result, the intrinsic phonon-limited mobility of these strained $MoS_2$ transistors is improved by over a factor of 8, also corroborated by first-principles calculations of phonon-limited electron mobilities. Our proposed approach, involving surface nanotopography for fabricating strained $MoS_2$ transistors and the resultant performance enhancement engineered through the reduction of electron-phonon scattering, opens novel design and integration possibilities toward high-performance 2DM devices.

## Methods
### Material synthesis and exfoliation
Monolayer $MoS_2$ flakes were grown by MOCVD. A c-plane sapphire chip was selected as growth substrate, which was annealed in air for 6 h to get an atomic smooth surface and spin-coated with 0.026 mol/L NaCl solution in deionized water to suppress nuclear density and accelerate growth rate. The chip was then loaded into a tube furnace, where the temperature and gas flow rate can be controlled by LabView. During growth, molybdenum hexacarbonyl (Mo(CO)$_6$) and hydrogen sulfide (H$_2$S) was carried into the quartz tube as precursors by argon (Ar) with flow rates of 6 sccm and 3 sccm, respectively. The Mo(CO)$_6$ precursor was stored in a bubbler immersed in water bath whose temperature was kept at 15 °C to achieve a constant vapor rate. To obtain a monolayer by balancing the growth rate, small amounts of H$_2$ and O$_2$ were introduced separately into the growth chamber. The growth process lasted for 30 min at 850 °C with a pressure of 850 mbar. At the end of the growth, the precursor supply was abruptly cut-off and the furnace cooled down naturally to room temperature with a 200 sccm of Ar flow to remove gaseous residues. Monolayer $MoS_2$ flakes were also exfoliated from 2H-$MoS_2$ bulk crystal (HQ Graphene) onto PDMS substrates.

### Fabrication of grayscale dielectric nanopatterns
A 10 wt% solution of polyphthalaldehyde (PPA, Allresist) in anisole (Sigma–Aldrich Chemie GmbH) was spin-coated at 5000 rpm on $SiO_2$ (200 nm thick thermally grown dry oxide)/Si (500 µm thick) substrate

and soft baked at 110 °C for 2 min. Grayscale nanostructures, biaxial sinusoidal wave $(f(x,y) = A[cos(gx) + cos(gy)])$, were patterned on PPA utilizing a commercial t-SPL system (Nanofrazor, Heidelberg Instruments Nano AG) and thermal cantilevers (NanoFrazor Monopede). Biaxial sinusoidal wave designs are converted into grayscale bitmaps consisting of $20 \times 20$ nm² pixel grids. The normalized depth was set to 256 levels. MATLAB (version R2020b) was utilized for grayscale bitmap generation. Then, the grayscale bitmap image was imported into the t-SPL software by assigning the minimum depth (white pixel) and maximum depth (black pixel) as 5 nm and 90 nm, respectively. The writing heater temperature for t-SPL was set to 1050 °C, and patterning was performed with a step size of 20 nm. A scan speed of 25 µs per pixel was used with a force pulse of 5 µs. The nanopatterns were transferred from PPA to thin films of SiO₂ by using a commercial inductively coupled plasma (ICP)-based reactive ion etching (RIE) system (Alcatel AMS 200 SE). In the dry etching process, $SF_6/C_4F_8$ plasma with a flow rate of 30/70 sccm at 0.015 mbar pressure was used. RF ICP power was set to 1500 W (13.56 MHz RF field). 15 W RF was applied at the bottom electrode through a blocking capacitor that allows to build up a constant DC bias voltage to attract the ions. The wafer was positioned on the bottom electrode and was gripped on that bottom electrode with electrostatic clamping (ESC). That bottom electrode was kept at 20 °C and thermal contact with wafer was ensured thanks to Helium backpressure through the ESC. After pattern transfer by dry etching, the substrates were cleaned with Piranha solution (3:1 mixture of $H_2SO_4(96\%):H_2O_2(30\%)$) for 10 min.

## Fabrication of strained 2D transistors

The monolayer MoS₂ flakes were picked up and then transferred onto patterned SiO₂ chips using the dry-transfer method. The 2D flakes were first etched into a suitable geometry covering the patterned substrate by using e-beam lithography and XeF₂ etching. A second step of e-beam lithography was used to make electrode patterns. O₂ plasma was used to clean the PMMA residues and etch the 2D flake exposed. Then, a layer composed of 2 nm/60 nm thick Ti/Pt was thermally evaporated for the electrodes. Finally, a lift-off process was performed in acetone to remove the PMMA layer. The fabricated devices were annealed in a tube furnace to remove polymer residues and strengthen the contact adhesion before characterization. The devices were supported by a boat-shape holder and loaded into the quartz tube. Then, the tube was sealed tightly and pumped down to a low pressure around $10^{-7}$ mbar before annealing started. Afterwards, the temperature ramped up to 200 °C with several steps (50 °C per step) and lasted for 6 h. When the annealing process was done, the temperature went down naturally to room temperature.

## Raman spectroscopy

Raman spectroscopy was performed to measure the Raman shift caused by strain. Raman spectra were collected using a confocal Raman microscope system (inVia Qontor, Renishaw) coupled with an Olympus inverted optical microscope, and using a laser source with an excitation wavelength of 532 nm. A low excitation laser power (84 µW) was used to avoid sample damage. Raman spectra were acquired in the range of 0 to 1800 cm⁻¹ with a 30 s exposure time and an average of three measurements.

## TERS

To prepare TERS probes, Si AFM cantilevers (Nanosensors, Switzerland) were first oxidized in a furnace (Carbolite Gero, UK) at 1000 °C for 23 h to increase the refractive index of the surface, followed by UV-ozone (Ossila, UK) cleaning for 1 h. The cleaned probes were placed in a thermal evaporation chamber of a N₂ glovebox (MBraun, Germany). The AFM cantilevers were coated with a 150 nm thick layer of Ag (Advent Research Materials, UK) at a rate of 0.5 nm/s under $10^{-7}$ mbar pressure. TERS measurements were performed under ambient

conditions using a side-illumination system consisting of a Raman spectrometer (HORIBA Scientific, France) and an atomic force microscope (AIST-NT, USA). 532 nm excitation laser was incident on the sample at an angle of 60° with respect to the surface and focused on the sample using a 100×, 0.7 NA objective lens (Mitutoyo, Japan). TERS line mapping was performed using a step size of 50 nm, spectrum acquisition time of 10 s and a laser power of ca. 260 µW at the sample. TERS spectra were collected using a spectrometer grating of 1800 lines/mm and a CCD detector.

## AFM and SEM analyses

AFM topography characterization of the fabricated grayscale nanostructures on SiO₂ thin films were performed with a Bruker FastScan AFM (ScanAsyst mode). The scanning-probe analysis software Gwyddion (version 2.59) was used for the purposes of data visualization and surface profile characterization. AFM images of the monolayer MoS₂ flake on the grayscale substrates were taken in PeakForce QNM® mode using the Multimode (Bruker) Scanhead and Nanoscope V controller (Bruker). ScanAsyst-Air cantilevers with a spring constant of 0.4 N/m were utilized, and peak forces were set to 10 nN for quantitative mechanical characterization. The devices were imaged using SEM (Zeiss MERLIN SEM) to analyze if the 2D flake follows the substrate.

## TEM analysis

A lamella of the cross-section device was prepared using focused ion beam (FIB) and SEM imaging. The target area was selected, and a carbon layer was deposited by electron beam-assisted (5 kV) and ion beam-assisted (30 kV, 150 pA) depositions. The former is used to protect MoS₂ from following ion implantation and surface damage. A lamella was cut perpendicular to the device surface to observe the interface between the MoS₂ and the patterned SiO₂. A FEI Talos TEM was used to study the interface between the MoS₂ and the SiO₂. The high-angular annular dark field (HAADF) STEM detector was used to image the sample with an accelerating voltage of 200 kV. EDX was then performed to analyze the material composition of the interface.

## Electrical measurement

Electrical measurements of all devices were performed at room temperature and ambient conditions. Some devices were also characterized in vacuum ($4 \times 10^{-6}$ mbar). For the Schottky barrier height extraction, some of the selected devices were measured for a set of temperatures ranging from 80 to 300 K, also in vacuum. The standard DC measurements were performed using a HP4156A Semiconductor Parameter Analyzer and a Cascade Summit probe station.

## DFT calculations

First-principles calculations are performed using the Quantum ESPRESSO (QE) distribution[69,70] in the framework of 2D density-functional and density-functional perturbation theory[60], including a cutoff on the Coulomb interaction to implement the correct 2D boundary conditions and gates to simulate electrostatic doping as induced in common FET setups. The exchange-correlation functional is approximated using the Perdew-Burke-Ernzerhof (PBE) formulation of the generalized-gradient approximation[71]. We explicitly include spin-orbit coupling in our simulations by using fully relativistic norm-conserving pseudopotentials[72] from the Pseudo-Dojo library[73] with a kinetic energy cutoff of 70 Ry. To ensure fine sampling close to the Fermi level, we use a non-uniform grid[14] with the equivalent of a $96 \times 96 \times 1$ grid for electron momenta and Fermi-Dirac smearing corresponding to room temperature (0.002 Ry). A denser grid of $120 \times 120 \times 1$ is used for the non-self-consistent calculation of the valley structure. Using the model of the Ref. 14 we corrected the results of a reference first-principles calculation for mobility by adjusting $\Delta E_{KQ}$ during post-processing, before solving the BTE to determine the

mobility. The model accounts for the fact that changes in $\Delta E_{KQ}$ modify energy selection rules for intervalley scattering, as well as the magnitude of intravalley electron-phonon coupling via a peculiar multivalley screening mechanism. Its parametrization relies on a single full first-principles calculation for the flat system, where $\Delta E_{KQ}$ is the sole external parameter[14]. $\Delta E_{KQ}$ as a function of strain is computed with DFT, as shown in Figs. 5d and 5e. The results are shifted by a constant such that the zero-strain value matches $\Delta E_{KQ}$ of 34 meV obtained using the experimental structure parameters[74]. See Supplementary Figs. 32–35 for further details.

## Data availability

All data that support the key findings in this study are available within the main text and the Supplementary Information file. All raw data generated during the current study are available from the corresponding authors upon request. The data used to produce the simulation results presented in this work are available at the Materials Cloud Archive (https://doi.org/10.24435/materialscloud:j5-7n).

## Code availability

The relevant implementation/code is already in Quantum ESPRESSO, as detailed in Ref. 60 Concerning the ab-initio model, the code and tools are available in Ref. 14, and the modifications introduced can be found at the Materials Cloud Archive (https://doi.org/10.24435/materialscloud:j5-7n).

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

## Acknowledgements

The authors thank the Center of Micro/Nanotechnology (CMi) of EPFL for the AFM facility support. X.L., B.E., A.C., G.B., and J.B. acknowledge funding from the European Research Council (ERC) under the European Union's Horizon 2020 research and innovation program (Project "MEMS 4.0", ERC-2016-ADG, grant agreement No. 742685), NFFA-Europe Pilot Project (the EU's H2020 framework program for research and innovation, grant agreement No. 101007417). X.L. acknowledges fundings from the National Natural Science Foundation of China (No. 62274013) and the National Key Research and Development Program of China (No. 2023YFB3405600). M.B. acknowledges the support of SNSF Eccellenza grant No. PCEGP2_194528, and support from the QuantERA II Programme that has received funding from the European Union's Horizon 2020 research and innovation program under Grant Agreement No 101017733. R.Z. acknowledges funding from the Swiss National Science Foundation through the NCCR MARVEL (project number 200021-143636) and the R'Equip program (grant number 206021-205312), as well as computational support from the Swiss National Supercomputing Centre. N.R. and N.M. acknowledge funding from the Swiss National Science Foundation (SNSF) and its National Centre of Competence in Research MARVEL on 'Computational Design and Discovery of Novel Materials' (grant number 182892). They also acknowledge computational support from the Swiss National Supercomputing Centre CSCS under project ID mr24. Z.W. and A.K. acknowledge financing from European Union's Horizon 2020 research and innovation program under grant agreements 881603 (Graphene Flagship Core 3) and No 964735 (EXTREME-IR). G.F. and M.P. received funding through Innosuisse (AFM with PORT: Atomic force microscope with photothermal off-resonance tapping, grant number 7879, ext ref. 36938.1 IP-EE), FNS (Video-rate

nanomechanical properties mapping using atomic force microscopy, grant number 7288, ext. ref. 200021_182562), H2020 (InCell - High speed AFM imaging of molecular processes inside living cellsInCell - High speed AFM imaging of molecular processes inside living cells, grant number 6823, ext ref. 773091, ETH Domain (An ecosystem for community driven scanning probe microscopy research and development, grant number 10126) and Eurostars (Correlated Analysis System for in-vivo Inspection of Semi-Conductor Process Wafers, grant number 9953, ext. ref. E!665 CAS-C).

## Author contributions

X.L., B.E., A.C., G.B., and J.B. conceived and designed the experiments for fabricating and characterizing strained 2D transistors. X.L., with the help of B.E. and A.C. and with the supervision of G.B. and J.B., performed preparation and transfer of 2D flakes, patterning/etching of 2D flakes and metal deposition/liftoff experiments. B.E., with the supervision of A.C., G.B., and J.B., performed t-SPL and dry etching. J.J. and M.B. helped transfer of 2D flakes and EBL. Z.W. and A.K. conceived and developed $MoS_2$ growth process. Z.W., with the supervision of A.K., performed $MoS_2$ growth. X.L., with the help of B.E. and A.C., performed Raman measurements and data analysis. X.L. performed SEM and TEM. X.L. performed electrical measurements and data analysis. M.P. and G.F. conceived and performed AFM on 2D transistors and data analysis. X.L., B.E., and J.J. helped AFM imaging. N.R. and T.S., with the supervision of N.M., performed the first-principles calculations. B.E., under the supervision of G.B., performed the calculation on electrical field. N.K. and S.B., with the supervision of R.Z. performed TERS measurement. X.L., B.E., N.M., G.B., and J.B. wrote the manuscript with input from all the authors. J.B. coordinated and supervised the research. All authors contributed to discussions regarding the research.

## Competing interests

The authors declare no competing interests.
