## [Peer Review File · Nature Communications]

Deterministic grayscale nanotopography to engineer mobilities in strained MoS₂ FETs

Corresponding Author: Professor Juergen Brugger

[The author's responses to all Reviewer comments can be found at the end of this file.]

Reviewer comments from the first round of review:

Reviewer #1

(Remarks to the Author)

The authors demonstrate enhanced 1L- MoS₂ FET performance (“8x improvement in on-state current”) when the 2D material is placed onto a sinusoidally templated substrate, where they find this improvement derives from tensile strain induced by substrate’s topography. Strain-enhanced mobility has already been explored experimentally and through simulations in 2D FETs [1-8], however, these other works only see a modest ~2x improvement in mobility and on-state current for 1L-MoS₂. The authors claim to observe more enhancement than these other works, for reasons that are not stated. The reviewer unfortunately cannot recommend the current version of the manuscript for publication, since major revisions are required (this includes reanalyzing data and distinguishing what is new in this work):

Major:

(1) Strain engineering is well known to control the bandgap in 2D TMDs, where tensile strain is generally understood to bring down the conduction band minimum. This means:

a. The authors absolutely need to compare device performance at a fixed gate overdrive ($V_G - V_T$) or gate-induced carrier density. The threshold voltage will vary from device-to-device, (unintentional) doping, bandgap reduction, etc, therefore it is unfair to compare I_{on} and field-effect mobility at an arbitrary maximum gate voltage or whenever transconductance is maximized.

- Note: This is especially important because there is a substantial threshold voltage shift in the strained structures versus flat structures (Fig. 4b shows a >10V difference by eye using CC method). Also, it’s not clear if each device is properly compared within the linear regime (Fig. 4c only shows $I_d V_d$ from 0-10 VG).

- Is there a trend in threshold voltage shift with applied strain?

b. The authors also need to reanalyze the contact resistance under fixed gate overdrive again. If there is no difference still even though there’s a reduction in bandgap, please explain. Why did the Schottky barrier measurements start at -15V instead of more negative (e.g., -30V) where Schottky barrier limits emission more? Device channel length is not stated (hopefully long channel devices were used).

(2) The authors look at field-effect mobility, which is a fair metric for comparing strain enhancement in FETs. However, field-effect mobility reflects transconductance and is well documented to over/underestimate the true channel mobility [9-12]. The reviewer would like the authors to compare their mobility values with other extraction methods [e.g., they will get RSHEET for free with their contact resistance analysis or use the Y-function method].

a. More on this point, the authors should have a statistical analysis for all their device metrics: I_{on} , FE mobility, R_c/R_{sh} extraction, V_T , etc. I_{on} and transconductance/mobility should be plotted versus channel length. Maybe plot these metrics versus gate overdrive. Please see <https://doi.org/10.1038/s41928-022-00798-8> for reference.

(3) The nature of the strain transfer is not entirely clear, nor do the authors confirm if there is unintentional doping from patterning the substrate. This is imperative since the authors presumably have biaxial strain in their simulations (not stated).

a. The authors say there is biaxial tensile strain, then show peak splitting in their in-plane phonon mode (consistent with uniaxial tensile strain from breaking crystal symmetry).

b. To corroborate Raman analysis, photoluminescence is usually done in conjunction. They can compare A-exciton shifts with DFT calculations. If they have biaxial strain, the A-exciton usually shifts ~100 meV per % (then ~40 meV per % for uniaxial).

c. Have the authors compared their A' peak positions and FWHMs in the strained and unstrained structures? This difference in peak position should help them untangle how much of the threshold voltage shift is from strain/doping.

d. Have the authors examined the strain transfer at the bottom of their sinusoidal structure (e.g., the valley instead of the hill)? Unclear if there is little-to-no strain there or if some compressive strain could be present. TERS/TEPL line scans would have

been fruitful.

(4) If the authors redo their analysis and still see >2x improvement from strain, then the authors should discuss why they observe more enhancement than other works.

a. What else is newly contributed from this work? Again, mobility enhancement from reduced intervalley scattering in strained 1L-MoS₂ is already documented.

b. Authors may want to consider benchmarking their devices against the literature to solidify the novelty of their work for the readers.

Minor:

(5) Have the authors conducted CV measurements on their patterned dielectrics?

(6) The authors say "edge-contacted S/D electrodes" in line 190, which could be misleading. Don't the authors have regular top contacts achieved using lift-off? Maybe the reviewer misunderstood.

(7) Authors state they fabricated and measured 400 devices, but do not show the statistics for all of them.

(8) There are also MOCVD and exfoliated devices, are they mixed in Fig. 4 or is this exfoliated alone? Maybe it would be best to separate them for transparency.

(9) "Mobility" needs to be explicitly stated as field-effect mobility in the figures.

(10) Some labels inside the figures say "strained" only instead of "X% strained" (e.g., Fig 4b,c, Fig. 5d).

(11) Fig. 5d should say E-E_c on the y-axis. Also, check MOCVD/exfoliated representative colors or labels in Fig. 5a,b,c (might be switched?)

(12) Gate leakage is not shown anywhere.

References

- [1] <https://doi.org/10.1021/acs.nanolett.2c01707>
- [2] <https://doi.org/10.1021/acsnano.3c10495>
- [3] <https://arxiv.org/abs/2309.10939>
- [4] <https://doi.org/10.1021/acsnano.3c03626>
- [5] <http://dx.doi.org/10.1088/0022-3727/48/37/375104>
- [6] <https://doi.org/10.1109/TED.2015.2461617>
- [7] <https://doi.org/10.1021/acs.nanolett.3c05091>
- [8] <https://doi.org/10.1021/acsnano.6b01149>
- [9] <https://doi.org/10.1063/1.4868536>
- [10] <https://doi.org/10.1002/sml.202100940>
- [11] <https://doi.org/10.1002/adma.201806020>
- [12] <https://doi.org/10.1038/ncomms10908>

Reviewer #2

(Remarks to the Author)

Liu and coworkers reported a new strain engineering approach to apply biaxial tensile strain to MoS₂ and investigated how it impacts the electrical transport properties of MoS₂ FETs made out of it. I found this work well-presented and the strain engineering technique they introduced interesting. I believe it is appealing to a broad audience working on strain engineering of 2D materials and 2D materials-based FETs. However, the below comments and questions should be addressed before the manuscript can be considered for publication.

1. It is not the first time strain has been applied to 2D materials by conforming 2D flakes onto patterned substrate. For example, Hinnefeld, J.H., Gill, S.T. & Mason, N. Graphene transport mediated by micropatterned substrates. *Appl. Phys. Lett.* 112, 173504 (2018). The authors should explain what is the advantage of using the sinusoidal wave pattern and why not using other periodic patterns other than sinusoidal? Besides the amplitude, does the wavelength of the pattern impact the strain transfer and the mobility?

2. The strained devices and the unstrained devices in this work have different gate and channel structure (sinusoidal patterned gate vs flat gate; sinusoidal-shaped channel vs straight channel). Both the gate and the channel structure may significantly impact the electrical behavior of the device. The authors briefly addressed this by claiming the difference of the gate capacitance is small. The authors should give a more detailed explanation of how they calculated the gate capacitance and how they define the channel length.

Also, the claim that 'the capacitance of the strained FET is also same as that of the flat one' in Supplementary Section 7 is not valid and contradicts the simulation result in Fig. S24.

3. It is well-known that the strain may adjust the bandgap of MoS₂. Could the authors comment on what is the impact of the bandgap of MoS₂ in their strained devices?

4. In the literature review section for recent strain engineering techniques of 2D materials, some popular strain engineering methods are omitted. For example, using thin film stressors (A. Azizimanesh, T. Peña, A. Sewaket, W. Hou, and S. M. Wu, *Appl. Phys. Lett.* 118, 213104 (2021); W. Hou, A. Azizimanesh, A. Dey, Y. Yang, W. Wang, C. Shao, H. Wu, H. Askari, S. Singh, S. M. Wu, *Nature Electronics* 7, 8 (2024)). The authors may consider having a more thorough literature review.

Reviewer #3

(Remarks to the Author)

In this manuscript, the authors developed a sophisticated nanofabrication for mobility enhancement in 2DM transistors, with the biaxial tensile strain induced by contact transfer of the 2DM layer on a grayscale nanopatterned SiO₂ dielectric. The SiO₂ pattern was fabricated by a novel technique of t-SPL and dry etching, which can be integrated into the existing silicon-based technology.

Under the tensile strain, the monolayer MoS₂ FETs exhibited a noticeable increase of electron mobility, exceeding 180 cm²/Vs at room temperature and improved by over a factor of 8 while comparing the unstrained (planar) devices.

Additionally, the first-principle calculations are conducted, to analyze and validate the mobility improvement, with respect to the intervalley scattering and coupling.

The nanotechnology and fabrication are intriguing, and demonstrates their superior mobility and electrical characteristics. Based on these points, I am positive in its possible publication in nature communications, after sufficiently addressing the following questions:

Comment 1. Strain engineering techniques in MoS₂ and 2DM shall be shortly stated in the introduction.

Comment 2. Although the AFM image shows the adhesion map and presence of the strained 2DM on the dielectric surface (Fig. 2i), the transfer quality with respect to the sheet size, defect, crack or wrinkle etc. needs to be included.

Comment 3. With a 65 nm peak-to-peak amplitude of the SiO₂ pattern, a 1% strain which corresponds to a shift of the E_{12g} peak by -4.4 cm⁻¹, was measured and extracted, rather than a 2.8% strain as theoretically expected. This large discrepancy (the 2-3 times lower than the maximum achievable limit) shall be clarified by the authors, regarding the SiO₂ pattern and the 2DM layer map.

Whether the 2DM layer is domestically, effectively adhered to surface of the patterned SiO₂ dielectric? Can the sinusoidal topography of 2DM layer be provided ?

Comment 4. Hysteresis-free transfer characteristics (Fig. 4d) have been observed in the strained MoS₂ transistor, low-frequency noise characterization can be included to identify the semiconducting layer and interface quality, as well.

Comment 5. The authors state that the electron mobility enhancement in the monolayer MoS₂ transistor, from 60 cm²/Vs measured in the unstrained to 150 cm²/Vs in the 0.65%-strain and 180 cm²/Vs in the 0.90%-strain, because of the drastic reduction in intervalley scattering due to the induced strain.

However, as theoretically predicted, the mobility of the single-layer MoS₂ is calculated to be reached up ~410 cm²/Vs , where the intrinsic phonon scattering is dominated (10.1103/PhysRevB.85.115317). As well, the experimental value of ~200 cm²/Vs has been reported for the single-layer MoS₂, while considering additional scattering mechanisms, e.g. the non-intrinsic Coulomb scattering.

In this manuscript, considering of the t-SPL and etching process to fabricate the gray-tone topography, the surface topography of the biaxial sinusoidal pattern and their impact on intrinsic and non-intrinsic scattering and mobility, shall be discussed and clarified, from the aspects of experiments and simulations.

Reviewer comments from the second round of review:

Reviewer #1

(Remarks to the Author)

The authors have thoroughly responded to my comments and reanalyzed their data under fixed overdrive. More importantly, they demonstrated that the novelty of their work is (1) their approach and (2) examining how impurities/defects play a role in mobility enhancement.

Reviewer #2

(Remarks to the Author)

The authors have satisfactorily addressed all my comments. The manuscript is now ready for publication.

Reviewer #3

(Remarks to the Author)

Additional concern, for the strained and flat FETs, the contact resistances extracted by TLM (Fig. S26) contradict the results of YFM (Fig. S27). Any explanation? the statistics shall be discussed.

Manuscript # NCOMMS-24-15521-T

Title: Deterministic grayscale nanotopography to engineer mobilities in strained MoS₂ FETs

Dear Editor and Reviewers,

We thank you and all the reviewers for their generally positive comments about our work and for transmitting to us their very useful suggestions, comments, and remarks that will allow us to improve the quality and clarity of our manuscript. Here you can find our point-to-point answers (Response and Revisions) to all reviewers' suggestions, comments, and remarks. The implemented changes are discussed below and highlighted in the revised manuscript (in red).

REVIEWER COMMENTS

Reviewer #1 (Remarks to the Author):

The authors demonstrate enhanced 1L- MoS₂ FET performance (“8x improvement in on-state current”) when the 2D material is placed onto a sinusoidally templated substrate, where they find this improvement derives from tensile strain induced by substrate’s topography. Strain-enhanced mobility has already been explored experimentally and through simulations in 2D FETs [1-8], however, these other works only see a modest ~2x improvement in mobility and on-state current for 1L-MoS₂. The authors claim to observe more enhancement than these other works, for reasons that are not stated. The reviewer unfortunately cannot recommend the current version of the manuscript for publication, since major revisions are required (this includes reanalyzing data and distinguishing what is new in this work):

Response:

We thank the reviewer for the comments. To highlight how our work compares with other published data, we have compiled a new overview table where we list the key parameters relevant to assess the performance of strained 2D transistors. In Supplementary Table S3, we summarize relevant published papers on strained MoS₂ FETs and extract the mobility enhancement at a given strain value. The mobility enhancement also depends on the carrier density and the strain type. The table shows that the reported values for mobility enhancements of monolayer MoS₂ FETs under strain vary substantially. Values include 1.7x, 1.8x, 2x, 2.52x, 20x, 60x, and 100x, respectively. The observed 8x improvement in our work is thus within the previously reported range. The mechanism behind the mobility change in our work can be explained by the substantial reduction of electron-phonon scattering under strain, as corroborated by the temperature-dependent electrical performance in Fig. 5. Furthermore, comprehensive first-principles calculations of electron-phonon coupling and mobilities were conducted and support the experimentally measured mobility improvements, which were also compared with published papers.

The recommended references [1-4] report the experimental results of strained MoS₂ FETs whereas the references [5-6] report the theoretical results. In addition to the suggested references from the reviewer, we also added and discussed other relevant literatures [R1-R4].

Reference [7] is not included in the comparison because it does not report relevant data on strain value, Raman shift and change in mobility. Reference [8] is excluded from the comparison because it reports a MoS₂ FET with a channel thickness of 7 nm, i.e. ~10 layers, which is significantly thicker than our monolayer devices.

We also appreciate the reviewer recommending an additional paper to guide our data interpretation (*Nature Electronics* 5, 416–423 (2022)). We have reanalyzed our data accordingly.

Newly added:

Supplementary Table S3: Comparison of performance of strained MoS₂ FETs

Refs	Material Thickness Method	Strain type	Strain value [%]	Raman shift [cm ⁻¹]	Carrier density [cm ⁻²]	Strained FET		Unstrained FET [cm ² V ⁻¹ s ⁻¹]
						Mobility [cm ² V ⁻¹ s ⁻¹]	Enhancement	
Experimental results								
[1]	MoS ₂ 1L CVD grown	Uniaxial tensile strain	0 ~ 0.7	E ¹ _{2g} : 1.61	1.1×10 ¹³	14 for 1L MoS ₂ at 0.7% strain	2x	7
[2]	MoS ₂ 1L CVD grown	Uniform biaxial tensile strain	0.7	E ¹ _{2g} : 3.3	NA	15.94	1.8x	8.71
[3]	MoS ₂ 1L CVD grown	Biaxial tensile strain	0.23	NA	NA	9.1 for 1L MoS ₂ at 0.23% strain	1.7x	5.4
[4]	MoS ₂ 1L and 2L Exfoliated	Uniaxial tensile strain	0.73-1.7	E ¹ _{2g} : 3.36-7.82	2×10 ¹²	127 for 1L MoS ₂ at 0.87% strain 82 for 2L MoS ₂ at 1.36% strain	2.52x (1L) 1.64x (2L)	50.5
[R1]	MoS ₂ 1L and 3L Exfoliated	Biaxial tensile strain	NA	0	9×10 ¹²	448 for 1L MoS ₂ 900 for 3L MoS ₂	20x (1L) 100x (3L)	20 (1L) 9 (3L)
[R2]	MoS ₂ 1L CVD grown	Local biaxial tensile strain	0.2-1.3	E ¹ _{2g} : ~6.5 A ¹ _{1g} : ~3.5	(0.8-1.8) ×10 ¹²	32	60x	0.5
[R3]	MoS ₂ 1L Exfoliated	Tensile strain	~0.7	NA	7.8×10 ¹²	~1150 for 1L MoS ₂ at 0.7% strain	100x	12.5
Our work	MoS ₂ 1L Exfoliated and CVD grown	Multiaxial tensile strain	0 ~ 1	E ¹ _{2g} : 4.4	3.7×10 ¹²	185 for 1L MoS ₂ at 0.9% strain	8x	23
Theoretical calculation results								
[5]	MoS ₂ 1L	Biaxial tensile strain	5	NA	1×10 ¹²	45	1.53x	30
[6]	MoS ₂ 1L	Biaxial tensile strain	0.4	NA	1×10 ¹²	NA	1.05x	NA
[R4]	MoS ₂ 1L	Uniaxial strain	2	NA	NA	0.9	6x	0.15
Our work	MoS ₂ 1L	Uniform tensile strain	0 ~ 1	NA	5×10 ¹²	485 for 1L MoS ₂ at 1% strain	8.46x	57.3

Detailed explanations are added below.

[1] Isha M. Datye, Alwin Daus, Ryan W. Grady, Kevin Brenner, Sam Vaziri, and Eric Pop. Strain-Enhanced Mobility of Monolayer MoS₂. *Nano Lett.* **22**, 8052–8059 (2022). <https://doi.org/10.1021/acs.nanolett.2c01707>

In this article, the authors reported mobility enhancement of about 2x for a 1.61 cm^{-1} redshift in E_{2g}^1 phonon, considering this shift as equivalent to 0.7% strain. However, in our work, we utilized the extensively documented and theoretically anticipated Raman peak redshift of $4.5 \text{ cm}^{-1}/\%$ strain for the E_{2g}^1 phonon as our benchmark. Therefore, a 1.5 cm^{-1} redshift is equivalent to 0.3% strain in our case, and we also report an enhancement of 2-3 for mobilities, as shown in Fig. 5a.

[2] Heechang Shin, Ajit Kumar Katiyar, Anh Tuan Hoang, Seok Min Yun, Beom Jin Kim, Gwanjin Lee, Youngjae Kim, JaeDong Lee, Hyunmin Kim, and Jong-Hyun Ahn. Nonconventional Strain Engineering for Uniform Biaxial Tensile Strain in MoS₂ Thin Film Transistors. *ACS Nano* **18**, 4414–4423 (2024). <https://doi.org/10.1021/acsnano.3c10495>

In this article, the authors reported lower enhancement factors for similar strains compared to our findings. The initial mobility reported in their MOCVD-grown MoS₂ was $\sim 8 \text{ cm}^2/\text{Vs}$, while it is $\sim 30 \text{ cm}^2/\text{Vs}$ in our case, which significantly depends on impurities and defects after the MOCVD process. Additionally, unlike our process, they deposited top-contact metal electrodes and gate dielectric on the MoS₂ layer to fabricate the final transistors. Furthermore, the stressed thin film SiO₂, which induces strain to the MoS₂ layer on top, is released in the final step of FET fabrication. Therefore, we cannot fairly compare the electrical performance of the final transistors with ours.

[3] Jerry A. Yang, Robert K. A. Bennett, Lauren Hoang, Zhepeng Zhang, Kamila J. Thompson, Antonios Michail, John Parthenios, Konstantinos Papagelis, Andrew J. Mannix, Eric Pop. Biaxial Tensile Strain Enhances Electron Mobility of Monolayer Transition Metal Dichalcogenides. <https://doi.org/10.48550/arXiv.2309.10939>

In this article, the authors reported mobility enhancement of almost 2x for 0.2% strain, which is very similar to what we report in our work.

[4] Yang Chen, Donglin Lu, Lingan Kong, Quanyang Tao, Likuan Ma, Liting Liu, Zheyi Lu, Zhiwei Li, Ruixia Wu, Xidong Duan, Lei Liao, and Yuan Liu. Mobility Enhancement of Strained MoS₂ Transistor on Flat Substrate. *ACS Nano* **17**, 14954–14962 (2023). <https://doi.org/10.1021/acsnano.3c03626>

The initial mobility reported in this article is $\sim 50 \text{ cm}^2/\text{Vs}$, and mobilities in strained transistors reach $>120 \text{ cm}^2/\text{Vs}$, which represents approximately a 2.5x enhancement. When starting with lower initial mobilities, the value can reach its limit of $\sim 120 \text{ cm}^2/\text{Vs}$ but provides a higher enhancement factor. Moreover, as shown in Fig. 5f, impurity concentration in monolayer MoS₂ significantly affects the maximum achievable mobility according to our theoretical demonstrations. Another important difference is gate capacitance, which affects doping concentrations. The reported gate capacitance in this article is $1.15 \times 10^{-8} \text{ F/cm}^2$, while it is $2.56 \times 10^{-8} \text{ F/cm}^2$ in our case. As shown in our Fig. 5b and 5f, we also reported 3-4x enhancement for low doping concentrations.

[5] Manouchehr Hosseini, Mohammad Elahi, Mahdi Pourfath, and David Esseni. Strain Induced Mobility Modulation in Single-Layer MoS₂. *J. Phys. D: Appl. Phys.* **48**, 375104 (2015). <http://dx.doi.org/10.1088/0022-3727/48/37/375104>

In this study, the authors reported an enhancement factor of $\sim 2x$ for a carrier concentration of 10^{11} cm^{-2} and a charged impurity concentration of 10^{12} cm^{-2} . In our work, as shown in Fig. 5f, we also report a very similar enhancement factor under very similar conditions: $N_d = 5 \times 10^{12} \text{ cm}^{-2}$ and $N_i = 1 \times 10^{12} \text{ cm}^{-2}$. Depending on the impurity level, we showed that different enhancement factors can be obtained.

[6] Manouchehr Hosseini, Mohammad Elahi, Mahdi Pourfath. Strain-Induced Modulation of Electron Mobility in Single-Layer Transition Metal Dichalcogenides MX₂ (M = Mo, W; X = S, Se). *IEEE Transactions on Electron Devices* **62**, 3192 – 3198 (2015). <https://doi.org/10.1109/TED.2015.2461617>

Very similarly to Ref. [5], they predicted an enhancement factor for impurity concentration of 10^{12} cm^{-2} .

[R1] Hong Kuan Ng, Du Xiang, Ady Suwardi, Guangwei Hu, Ke Yang, Yunshan Zhao, Tao Liu, Zhonghan Cao, Huajun Liu, Shisheng Li, Jing Cao, Qiang Zhu, Zhaogang Dong, Chee Kiang Ivan Tan, Dongzhi Chi, Cheng-Wei Qiu, Kedar Hippalgaonkar, Goki Eda, Ming Yang, and Jing Wu. Improving Carrier Mobility in Two-Dimensional Semiconductors with Rippled Materials. *Nat. Electron* **5**, 489–496 (2022)

[R2] Arijit Kayal, Sraboni Dey, Harikrishnan G., Renjith Nadarajan, Shashwata Chattopadhyay, and Joy Mitra. Mobility Enhancement in CVD-Grown Monolayer MoS₂ Via Patterned Substrate-Induced Nonuniform Straining. *Nano Lett.* **23**, 6629–6636 (2023)

[R3] Tao Liu, Song Liu, Kun-Hua Tu, Henrik Schmidt, Leiqiang Chu, Du Xiang, Jens Martin, Goki Eda, Caroline A. Ross, and Slaven Garaj. Crested Two-Dimensional Transistors. *Nat. Nanotechnol.* **14**, 223–226 (2019)

In summary, the original contributions of our work compared to the prior state of research are:

We developed a novel approach for fabricating **permanently strained** 2DM transistors at the nanoscale using a **deterministic gray-tone topography** patterning of gate dielectric for device integration purposes. This deterministic grayscale nanopatterning is achieved by thermal scanning probe lithography (t-SPL) with single-digit nanometer resolution, which cannot be achieved through other grayscale nanopatterning techniques such as E-beam lithography and interference lithography. This is an essential technology step towards the reliable fabrication and integration of compact nanoscale 2DM transistors with controlled strain.

- The surface of the gate dielectric patterned with sinusoidal wave topographies **induces localized tensile strain** in the monolayer MoS₂ flake transferred on top.
- The precise depth-to-pitch ratio control of nanotopographies provides **deterministic control of strain induced in 2DMs**.
- The newly developed nanopatterning allows introducing **strain gradients within the same 2DM flake** by adjusting the amplitude and spatial frequency of the sinusoidal waves in a controllable way.
- Grayscale nanotopographies provides improved **conformal attachment of 2DMs** by reducing wrinkling and free-standing of these atomically thin materials. This results in **improved dielectric-semiconductor interfaces** and a **mechanically stable** environment for further fabrication processes (e.g., resist coating for 2DM patterning, dielectric deposition, and lift-off metallization).
- Unlike sharp crested patterns, grayscale smooth nanopatterns results in a more **homogeneous distribution of strain** while keeping the strain **localized within the pattern area**.
- Our experimental findings are in good agreement with **theoretical DFT modeling** based on the band structure of strained MoS₂.

Revisions:

- We added Supplementary Table S3 showing a detailed comparison of several key features of strained MoS₂ FETs.
- In Lines 78-96 of the Introduction section, we added sentences to clarify the novelty of our work as follows: “The t-SPL with a lateral resolution below 10 nm and sub-nanometer vertical depth control enables fabrication of high-resolution grayscale nanopattern with deterministic aspect ratio control.^{45,47} The tensile strain in the 2DM is induced through the elongation of the 2DM during the process of contact-transferring a planar 2DM flake using an elastomeric stamp onto a grayscale sinusoidal silicon dioxide (SiO₂) dielectric. This sinusoidal topography is previously fabricated by t-SPL and dry etching and can be programmed by adjusting the aspect ratios, also referred to as depth-to-pitch ratios. The depth-to-pitch ratio control capability of t-SPL, which cannot be so precisely achieved through other grayscale nanopatterning techniques such as electron beam lithography⁴⁸ and interference lithography,⁴⁹ provides deterministic control of strain induced in 2DMs. Varying the depth-to-pitch ratios of nanotopographies allows for the introduction of areas with strain gradients within a single 2DM flake by adjusting the amplitude and spatial frequency of the sinusoidal waves. Compared to other approaches such as nanopillar arrays and rippled or crested substrates, grayscale nanotopographies also offer improved conformal attachment of 2DMs by reducing wrinkles and suspended parts. This results in improved dielectric-semiconductor interfaces and a mechanically more stable environment compatible with subsequent fabrication processes. In contrast to the 2DMs strained by sharp crested patterns, where strain is non-uniform and is very high at peaks and very low on flat parts, grayscale nanopatterns offer a more homogeneous distribution of strain while still keeping the strain localized within the pattern area.”.
- In Lines 413-415 of the Conclusion section, we added one sentence about the novelty of our work as follows: “Precisely patterned surface topography at the single-digit nanometer scale enables deterministic changes in the tensile strain induced in 2D materials, thereby locally altering their electrical and optical properties while offering a seamless device integration option.”

Major:

(1) Strain engineering is well known to control the bandgap in 2D TMDs, where tensile strain is generally understood to bring down the conduction band minimum. This means:

a. The authors absolutely need to compare device performance at a fixed gate overdrive ($V_G - V_T$) or gate-induced carrier density. The threshold voltage will vary from device-to-device, (unintentional) doping, bandgap reduction, etc, therefore it is unfair to compare I_{on} and field-effect mobility at an arbitrary maximum gate voltage or whenever transconductance is maximized.

- Note: This is especially important because there is a substantial threshold voltage shift in the strained structures versus flat structures (Fig. 4b shows a >10V difference by eye using CC method). Also, it's not clear if each device is properly compared within the linear regime (Fig. 4c only shows $I_d V_d$ from 0-10 VG).

Response:

We acknowledge that it is not appropriate to compare the device performance (I_{on} and μ_{FE}) at different gate overdrive. In our originally submitted manuscript, we extracted the average value of the transconductance (dI_D/dV_{GS}) in the linear region of I_D - V_{GS} . In the revised version, we reanalyzed the metrics according to the suggested and added paper (*Nature Electronics* 5, 416–423 (2022)).

We added plots of field-effect mobility as a function of gate-induced carrier density (n_s) and compared the field-effect mobilities of the strained and flat FETs at a fixed n_s .

Regarding the threshold voltage (V_T), it is true that V_T varies from device to device. In the revised version, we have analyzed the V_T of the strained and flat FETs as a function of the channel length. V_T is estimated using the linear extrapolation method.

In the revised version, the range of output curves (I_D - V_{DS}) is extended to from -3 V to 3 V covering both the linear and the saturation regime.

Revisions:

- In Lines 269-280, a paragraph was added as “The threshold voltage (V_T) of the strained transistors is lower than the flat ones (Fig. 4e and Supplementary S20), suggesting that the electron density increases with the decreasing band gap under tensile strain. To account for shifts in V_T , we compared transistor metrics, i.e., on-state current (I_{on}), contact resistance (R_c) and field-effect mobility (Fig. 4f-h) at the same carrier density calculated using $n_s = C_{ox}(V_{GS} - V_T)/q$ where C_{ox} is the gate dielectric capacitance and q is the elementary charge. The I_{on} of strained and flat transistors linearly increases with the carrier density. The R_c of both strained and unstrained FETs are less than $2 \text{ k}\Omega\cdot\mu\text{m}$ at $n_s \geq 4 \times 10^{12} \text{ cm}^{-2}$, indicating that the devices are channel-dominated (Fig. 4g and Supplementary Fig. S26). For a fair comparison, the carrier-density-dependent field-effect mobility of both transistors was compared at the same carrier density (Fig. 4h). We also compared the transconductance of both transistors for different channel lengths (Fig. 4j). Key parameters of the strained and unstrained FETs are shown in Supplementary Table S2 and Table S3.⁵³”.

Ref [53]: Cheng, Z. et al. How to report and benchmark emerging field-effect transistors. *Nat. Electron.* 5, 416–423 (2022).

- We added the Supplementary Table S2 in SI.

Supplementary Table S2: Key parameters of the strained and unstrained FETs

Strain value	L_{ch} [μm]	I_{on} [$\mu\text{A}/\mu\text{m}$]	V_T [V]	R_{sh} [$\text{k}\Omega/\square$]	R_c (YFM) [$\text{k}\Omega\mu\text{m}$]	R_c (TLM) [$\text{k}\Omega\mu\text{m}$]	μ_{FE} [$\text{cm}^2\text{V}^{-1}\text{s}^{-1}$]	μ_{con} [$\text{cm}^2\text{V}^{-1}\text{s}^{-1}$]
0.90%	1	154	27	7.4	1.2	1.4	185	166
0.65%	1	116	28	8.9	1.5	1.8	154	146
0.36%	1	87	30	12.4	2.4	2.1	101	109
0%	1	32	34	66	3.6	1.2	26	23

- Figure 4 was revised by adding six panels (Fig. 4e-4j) and adding more data in Fig. 4b and 4c.
 - Fig. 4h was added with the plots of field-effect mobility of the strained and flat FETs as a function of gate-induced carrier density (n_s).
 - Fig. 4e was added with the V_T of the strained and flat FETs as a function of the channel length.
 - Fig. 4c was added with a wide range of V_{DS} from -3 V to 3 V.

Fig. 4 | Electrical characteristics of the strained MOCVD grown monolayer MoS₂ transistors. **a**, Schematic cross-section and electrical connections of a back-gate strained FET. The structure comprises the heavily p-type doped Si substrate as global back-gate, and a dry thermally grown SiO₂ as back-gate dielectric. **b**, Room-temperature transfer curve of the strained FET measured at drain-to-source bias voltage, $V_{DS} = 1$ V. **The gate leakage current (I_{GS}) versus V_{GS} plotted as well.** **c**, I_D as a function of V_{DS} at varying V_{GS} for strained (left) and unstrained (right) transistors fabricated on the same MoS₂ flake with the same channel length and width. **d**, Forward (red solid curve) and reverse (orange dash curve) transfer curves on logarithmic scales. The curves are hysteresis-free in the entire voltage range of the device after thermal annealing. **e**, **Threshold voltage (V_T) varying with channel length at $V_{DS} = 1$ V and $n_s = 4 \times 10^{12} \text{ cm}^{-2}$.** **f-h**, Transistor metrics (f: on-state current, I_{on} , g: contact resistance and h: field-effect mobility) of strained ones and flat ones are compared with the varying carrier density at $V_{DS} = 1$ V. **i-j**, I_{on} and transconductance (g_m) of both transistors are compared with varying channel length. The error bars in panels e-j represent the maximum and minimum values with the representative points corresponding to the average values for 120 transistors in panels e, i, and j, and for 30 transistors in panels f-h. **k**, Typical I_D - V_{GS} of strained FETs (red) and flat FET (blue) on SiO₂ dielectrics with various pattern amplitudes. **l**, Statistical distribution of electron mobility of the strained FETs on the substrate with different strain (or pattern amplitude).

- Is there a trend in threshold voltage shift with applied strain?

Response:

The threshold voltages (V_T) of the strained and flat devices were compared in the Fig. 4e and Fig. S20. The threshold voltage decreases with increased tensile strain.

Revisions:

- In Lines 269-271, we added a sentence related how strain affects V_T were added as “The threshold voltage (V_T) of the strained transistors is lower than of the flat ones (Fig. 4e and Supplementary Fig. S20), suggesting that the electron density increases with the decreasing bandgap under tensile strain.”.
- Fig. 4e was added with the V_T of the strained and flat FETs as a function of the channel length.
- A figure about V_T was added as Fig. S20 in SI.

Supplementary Fig. S20: Threshold voltage V_T as a function of applied biaxial strain from 0% (unstrained) to 1%. The error bars present the maximum and minimum values with the representative points corresponding to the average values.

b. The authors also need to reanalyze the contact resistance under fixed gate overdrive again. If there is no difference still even though there's a reduction in bandgap, please explain. Why did the Schottky barrier measurements start at -15V instead of more negative (e.g., -30V) where Schottky barrier limits emission more? Device channel length is not stated (hopefully long channel devices were used).

Response:

We compare the contact resistance calculated using different methods: transfer length method (TLM) and Y-function method (YFM) as shown in Supplementary Section 6.4. In the revised version, we added the result of R_c varying with carrier density and the Schottky barrier measurements starting from -40 V to 30 V in Fig. S17-19. The device channel length was 1 μm and has now been added in the captions of Fig. S23-25.

Revisions:

- In Lines 271-280, we added some sentences as follows “To account for shifts in V_T , we compared transistor metrics, i.e., on-state current (I_{on}), contact resistance (R_c) and field-effect mobility (Fig. 4f-h) at the same carrier density calculated using $n_s = C_{ox}(V_{GS} - V_T)/q$ where C_{ox} is the gate dielectric capacitance and q is the elementary charge. The I_{on} of strained and flat transistors linearly increases with the carrier density. The R_c of both strained and unstrained FETs are less than 2 $\text{k}\Omega \cdot \mu\text{m}$ at $n_s \geq 4 \times 10^{12} \text{ cm}^{-2}$, indicating that the devices are channel-dominated (Fig. 4g and Supplementary Fig. S26). For a fair comparison, the carrier-density-dependent field-effect mobility of both transistors was compared at the same carrier density (Fig. 4h). We also compared the transconductance of both transistors for different channel lengths (Fig. 4j). Key parameters of the strained and unstrained FETs are shown in Supplementary Table S2 and Table S3.⁵³”.

- Fig. 4g was added showing R_c as a function of carrier density and comparing the difference between strained and flat FETs.
- A subsection in SI was added showing the calculation of sheet resistance and contact resistance as below:

6.4 Analysis of Sheet Resistance and Contact Resistance

6.4.1 Transfer length method (TLM)

The total resistance per unit of channel width R_{tot} (in $\Omega \cdot m$) of the transistors in the TLM structure consists of the channel resistance per unit of channel width R_{ch} (in $\Omega \cdot m$) and the contact resistance per unit of channel width R_c (in $\Omega \cdot m$), is expressed as:

$$R_{tot} = R_{ch} + 2R_c = R_{sh} \cdot L_{ch} + 2R_c \quad (S4)$$

where R_{sh} (in Ω/\square) is the sheet resistance, L_{ch} is the channel length. The total resistance, R_{tot} can be plotted as a function of the channel length, L_{ch} . Therefore, by linearly fitting the curve, the sheet resistance and the contact resistance can be readily obtained from the slope of the fitting line and from the intercept with the vertical axis when $L_{ch} = 0$, respectively. Supplementary Fig. S26 shows the data.

Supplementary Fig. S26: Sheet resistance R_{sh} and contact resistance R_c extractions using the transfer-length method (TLM) when V_{GS} is 60 V.

6.4.2 Y-function method (YFM)

To estimate the contact resistance of the flat and strained MoS₂ FETs, we employ the Y-function method (YFM).⁵ According to YFM, the Y-function can be expressed as

$$Y = \frac{I_D}{\sqrt{g_m}} = \sqrt{\mu_0 C_{ox} V_{DS} \frac{W_{ch}}{L_{ch}}} (V_{GS} - V_T) \quad (S5)$$

where g_m is the transconductance, μ_0 is the low field mobility, C_{ox} is the gate dielectric capacitance, V_T is the threshold voltage, W_{ch} is the channel width, and L_{ch} is the channel length. In strong inversion, Y should be linear in gate voltage with intercept and slope giving the threshold voltage V_T , and the low field mobility parameter μ_0 , respectively. Here, V_T is extracted from the $I_D/g_m^{0.5}$ curve. Using linear fitting of the plot of Y-function with respect to V_{GS} , μ_0 can be extracted. The expression of mobility attenuation factor θ due to contact resistance, surface roughness, and phonon scattering as a function of V_{GS} can be written as

$$\theta = \frac{\left[\frac{I_D}{(g_m(V_{GS} - V_T))} - 1 \right]}{(V_{GS} - V_T)} \quad (S6)$$

The mobility attenuation factor includes the effects of the S/D series resistance and can be expressed as

$$\theta = \theta_0 + 2R_c \mu_0 C_{ox} \frac{W_{ch}}{L_{ch}}, \quad (S7)$$

where θ_0 is the intrinsic mobility attenuation factor.⁶ For 2D transistors, θ_0 is considered negligible. As a result, the contact resistance R_c can be calculated from θ ,

$$2R_c = \frac{\theta}{(\mu_0 C_{ox} \frac{W_{ch}}{L_{ch}})} \quad (S8)$$

In strong inversion, this function (θ versus V_{GS}) is expected to be a constant equal to the value of the mobility reduction coefficient. Using Eqs. (S5)-(S8), we estimate R_c of the strained and flat FETs.

Supplementary Figure S27: Calculation of contact resistance using the Y-function method (YFM). (a, c) Y function as a function of V_{GS} for the strained and flat FETs, respectively. (b, d) Mobility attenuation factor θ as a function of V_{GS} for the strained and flat FETs, respectively.

- Supplementary Figs. S23-25 were corrected with the V_{GS} starting at more negative voltage (-40V) as below.

Supplementary Fig. S23: Arrhenius plots at different values of V_{GS} in the strained FET with a channel length of 1 μm .

Supplementary Fig. S24: Arrhenius plots at different values of V_{GS} in the flat FET with a channel length of 1 μm .

Supplementary Fig. S25: Schottky barrier height (SBH) for (a) a strained transistor and (b) a flat transistor on the same flake. The extracted value of the SBH can change by about $\pm 15\%$ for a reasonable variation of the extraction region. Strained and flat FETs both have channel lengths of 1 μm .

(2) The authors look at field-effect mobility, which is a fair metric for comparing strain enhancement in FETs. However, field-effect mobility reflects transconductance and is well documented to over/underestimate the true channel mobility [9-12]. The reviewer would like the authors to compare their mobility values with other extraction methods [e.g., they will get RSHEET for free with their contact resistance analysis or use the Y-function method].

a. More on this point, the authors should have a statistical analysis for all their device metrics: I_{on} , FE mobility, R_c/R_{sh} extraction, V_T , etc. I_{on} and transconductance/mobility should be plotted versus channel length. Maybe plot these metrics versus gate overdrive. Please see <https://doi.org/10.1038/s41928-022-00798-8> for reference.

Response:

Thank you for this suggestion, which simplifies the performance comparison of the strained and unstrained FETs in terms of many critical device metrics. In the revised version, we added a table as Table S2 in SI. We compared the field-effect mobility ($\mu_{FE} = \frac{L_{ch}}{W_{ch}} \frac{1}{C_{ox}} \frac{1}{V_{DS}} \frac{dI_{DS}}{dV_{GS}}$) and the conductivity mobility ($\mu_{con} = \frac{1}{C_{ox}} \frac{1}{V_{GS} - V_T} \frac{1}{R_{sh}}$) in the Table S2, as well as other metrics, including I_{on} , V_T , R_c , and R_{sh} . In the revised main text, we added the field-effect mobility plotted versus carrier density (Fig. 4h) and transconductance plotted versus channel length (Fig. 4j), as well as I_{on} showing in Fig. 4f and Fig. 4i.

Revisions:

- We added a paragraph to introduce the calculation of the conductivity mobility in Section 6. A table of comparison of all the device metrics is added in SI as Table S2.

Conductivity mobility m_{con} is also used to estimate the carrier transport property. m_{con} has the advantage of reflecting the channel material properties and the quality of the channel-dielectric interface.¹ The conductivity mobility m_{con} is calculated using the equation

$$\mu_{con} = \frac{1}{qn_s R_{sh}} \quad (S2)$$

$$n_s = \frac{C_{ox}(V_{GS}-V_T)}{q} \quad (S3)$$

where n_s is the carrier density near the source and R_{sh} is extracted from the slope of the TLM plots.

Supplementary Table S2: Key parameters of the strained and unstrained FETs

Strain value	L_{ch} [μm]	I_{on} [$\mu\text{A}/\mu\text{m}$]	V_T [V]	R_{sh} [$\text{k}\Omega/\square$]	R_c (YFM) [$\text{k}\Omega\mu\text{m}$]	R_c (TLM) [$\text{k}\Omega\mu\text{m}$]	μ_{FE} [$\text{cm}^2\text{V}^{-1}\text{s}^{-1}$]	μ_{con} [$\text{cm}^2\text{V}^{-1}\text{s}^{-1}$]
0.90%	1	154	27	7.4	1.2	1.4	185	166
0.65%	1	116	28	8.9	1.5	1.8	154	146
0.36%	1	87	30	12.4	2.4	2.1	101	109
0%	1	32	34	66	3.6	1.2	26	23

- In Lines 276-280, we added some sentences as follows “For a fair comparison, the carrier-density-dependent field-effect mobility of both transistors was compared at the same carrier density (Fig. 4h). We also compared the transconductance of both transistors for different channel lengths (Fig. 4j). Key features of the strained and unstrained FETs are shown in Supplementary Table S2 and Table S3.⁵³”.
- In the revised Fig. 4, we added six panels analyzing V_T , I_{on} , R_c , field-effect mobility and transconductance versus carrier density and channel length.

(3) The nature of the strain transfer is not entirely clear, nor do the authors confirm if there is unintentional doping from patterning the substrate. This is imperative since the authors presumably have biaxial strain in their simulations (not stated).

Response:

The tensile strain is induced through the elongation of the 2DM during the process of contact-transferring the planar 2DM onto the silicon dioxide (SiO_2), that is previously patterned into the sinusoidal topographic surface by t-SPL and dry etching. Figure S14 shows schematically how the strain is introduced. Figure S7 shows the step-by-step process flow of transferring MOCVD grown 2D flakes onto locally patterned substrates. After the transfer, as explained in detail in this letter, the main text, and the SI document, we observe an almost homogeneous distribution of strain while keeping the strain localized within the pattern area, according to local strain characterization performed by TERS. As the strain results from elongation, higher depths in sinusoidal nanopatterns result in higher tensile strain in 2DMs.

The doping in the channel that derives from electrostatic gating and also substrate scattering has been taken into account already in the original version. Since the wavy substrate and the flat substrate have similar surface quality, we consider that unintentional doping contributes similarly to the carrier transport of the transistors.

In our first-principles simulations, we explored the impact of applying uniform tensile strain across the material, bypassing the direct simulation of intricate topographical features due to their computational complexity at the atomistic level. These features demand prohibitively large computational resources, making it impractical to replicate realistic patterns. Our focus lies on analyzing the effects of this constant strain, particularly its influence

on intrinsic scattering mechanisms and electron mobility. In the revised version, we stated the nature of simulated strain to make this clearer.

Revisions:

- In Lines 78-96 of the Introduction section, we added sentences to clarify the novelty of our work as follows: “The t-SPL with a lateral resolution below 10 nm with sub-nanometer vertical depth control enables fabrication of high-resolution grayscale nanopattern with deterministic aspect ratio control.^{45,47} The tensile strain in the 2DM is induced through the elongation of the 2DM during the process of contact-transferring a planar 2DM flake using an elastomeric stamp onto a grayscale sinusoidal silicon dioxide (SiO₂) dielectric. This sinusoidal topography is previously fabricated by t-SPL and dry etching and can be programmed by adjusting the aspect ratios, also referred to as depth-to-pitch ratios. The depth-to-pitch ratio control capability of t-SPL, which cannot be so precisely achieved through other grayscale nanopatterning techniques such as electron beam lithography⁴⁸ and interference lithography,⁴⁹ provides deterministic control of strain induced in 2DMs. Varying the depth-to-pitch ratios of nanopatterns allows for the introduction of areas with strain gradients within a single 2DM flake by adjusting the amplitude and spatial frequency of the sinusoidal waves. Compared to other approaches such as nanopillar arrays and rippled or crested substrates, grayscale nanopatterns offer also improved conformal attachment of 2DMs by reducing wrinkles and suspended parts. This results in improved dielectric-semiconductor interfaces and a mechanically more stable environment compatible with subsequent fabrication processes. In contrast to the 2DM strained by sharp crested patterns, where strain is non-uniform and is very high at peaks and very low on flat parts, grayscale nanopatterns offer a more homogeneous distribution of strain while still keeping the strain localized within the pattern area.”.
- For the first-principles simulations, in Lines 338-339, we added “... by applying uniform tensile strain across the material”.
- We added the Supplementary Fig. S7 in SI to clarify the transfer process of 2D flakes onto the sinusoidal substrates.

Supplementary Fig. S7: Pick-up and transfer of MoS₂ flakes onto a locally patterned substrate. (a) A polycarbonate (PC) film is mounted on top of a PDMS layer with a curved surface. The PC film gets in contact with flakes and the contact area expands with the stage temperature increasing by 95 °C. (b) The PC film is detached from the substrate and several flakes are peeled off. (c) The PC film with the flakes is transferred onto a target substrate and the PDMS layer is detached at 180 °C. The PC film in contact with the substrate stays on the substrate together with the flakes. (d) The PC film is dissolved in chloroform for 1 hour. The MoS₂ flake is successfully transferred on the target patterned substrate.

a. The authors say there is biaxial tensile strain, then show peak splitting in their in-plane phonon mode (consistent with uniaxial tensile strain from breaking crystal symmetry).

Response: Thank you for this very interesting point and observation. We reevaluated the nature of the strain applied in our experiments and have made some clarifications and updates. Initially, we described the strain as

biaxial, considering previous publications (such as <https://www.nature.com/articles/ncomms8381>). However, considering the reviewer's observation and a thorough reanalysis of our sinusoidal wave design modulated in both the x- and y-directions, we found that the strain applied is actually more complex. During contact transfer, 2D materials initially make contact with the peaks of the nanopatterns and then begin to elongate. Due to the design of the sinusoidal waves, the strain distribution exhibits multiaxial characteristics with varying degrees of elongation in different directions. We have updated our manuscript using the term "multiaxial strain" to more accurately describe the nature of the strain. This terminology accounts for the overall multiaxial strain affecting the MoS₂ crystal symmetry, and therefore this peak splitting effect is due to multi-directional elongations.

The peak splitting of the Raman peak E_{2g}¹ was observed using both micro-Raman and AFM-based tip-enhanced Raman spectroscopy (AFM-TERS) techniques. Fig. 3 shows the strain characterization of the strained FETs. Fig. 3a shows the splitting of Raman peak E_{2g}¹ of strained MoS₂ and flat MoS₂. In Fig. 3f, the splitting of the E_{2g}¹ peaks is also observed using AFM-TERS at the spatial resolution of 50 nm, consistent with the results obtained from micro-Raman measurements. This indicates that both techniques, micro-Raman and AFM-TERS, measure a similar level of strain in the MoS₂ layer, and the similar strain level is distributed in the entire sinusoidal nanopattern. Additional TERS measurements from other patterned structures are shown in Supplementary Fig. S18.

Revisions:

In Lines 74 and 115, "biaxial" words were replaced with "multiaxial".

b. To corroborate Raman analysis, photoluminescence is usually done in conjunction. They can compare A-exciton shifts with DFT calculations. If they have biaxial strain, the A-exciton usually shifts ~100 meV per % (then ~40 meV per % for uniaxial).

Response:

In the revised version, we have added Supplementary Fig. S19 for the experimental results of bandgap modulation obtained through photoluminescence (PL) measurements and its corresponding micro-Raman spectroscopy characterization on sinusoidal nanopatterns. We observe a ~38 meV shift for a 1.3 cm⁻¹ shift in the E_{2g}¹ peak position, corresponding to ~132 meV/% strain and we have updated Fig 5e caption where we predict ~150 meV/% strain from our first-principles simulations.

Revisions:

- In Lines 359-360, we added a sentence "The energy separation variation rate is around 150 meV/% strain, close to the experimental value obtained from the photoluminescence measurement (Supplementary S19).".
- In Lines 399-400 of the legend of Fig. 5e, we added a sentence "See Supplementary Fig. S19 for the experimental results on the strain induced bandgap modulation obtained through photoluminescence (PL) measurements.".
- We added the Supplementary Fig. S19 in SI to analyze the shift rate of A-exciton.

Supplementary Fig. S19: (a) Optical image of the exfoliated MoS₂ flake transferred onto a nanoengineered substrate. (b) Micro-Raman spectroscopy characterization of strained 1L MoS₂ on sinusoidal nanopattern modulated in two dimensions and its (c) photoluminescence (PL) characterization to visualize bandgap

modulation of MoS₂ monolayer under biaxial tensile strain. We observe a ~38 meV shift for a 1.3 cm⁻¹ shift in the E¹_{2g} peak position, corresponding to ~132 meV/% strain.

c. Have the authors compared their A' peak positions and FWHMs in the strained and unstrained structures? This difference in peak position should help them untangle how much of the threshold voltage shift is from strain/doping.

Response:

We are not sure to fully understand this comment but provide an answer to the best of our comprehension, hoping that it helps to clarify it.

In the revised version, we added the Supplementary Table S1 comparing the A_{1g} positions and FWHMs of strained and unstrained MoS₂ FETs. We did not observe any significant shifts (0.2-0.4 cm⁻¹ that is much lower than the resolution) in out-of-plane vibration A' (also called A_{1g}) peak position. Additionally, we observed an increase in the A_{1g} FWHM up to 12% (TERS measurement) and 67% extracted from micro-Raman measurement, which is related to the induced strain. It is important to note that when evaluating the results, the micro-Raman laser spot size (~1 μm) is larger than some of the strained FET channels (~0.4 μm).

Revisions:

- In Lines 190-191, we added two sentences “However, no shift in Raman peak A_{1g} is observed. An increase in the A_{1g} FWHM of MoS₂ also indicates the effect of strain³¹ as compared in Supplementary Table S1.”
- The reference (Ref 31) was added “Li, Z. *et al.* Efficient strain modulation of 2D materials via polymer encapsulation. *Nat. Commun.* **11**, 1151 (2020).”.
- A table was added as Supplementary Table S1.

Supplementary Table S1: Information from Raman spectra of flat MoS₂ and strained MoS₂ FETs

Sample	A _{1g} position, cm ⁻¹ (micro-Raman)	A _{1g} FWHM, cm ⁻¹ (micro-Raman)	A _{1g} position, cm ⁻¹ (TERS)	A _{1g} FWHM, cm ⁻¹ (TERS)
Strained MoS ₂	404.5	9.72	406.7 (top)	7.03 (top)
			406.8 (slope)	7.52 (slope)
			406.6 (bottom)	7.35 (bottom)
Flat MoS ₂	404.3	5.81	406.4	6.74

d. Have the authors examined the strain transfer at the bottom of their sinusoidal structure (e.g., the valley instead of the hill)? Unclear if there is little-to-no strain there or if some compressive strain could be present. TERS/TEPL line scans would have been fruitful.

Response:

The TERS line scans were conducted on the sinusoidal structure in the direction perpendicular to the S/D electrodes, as shown in Fig. 3d. The bottom, slope and top positions were illustrated in Fig. 3e. Fig. 3f shows the TERS spectra of the three positions, each computed from 4 pixels covering an area of 100 × 100 nm² (Fig. 3e) of the patterned MoS₂, respectively. The E¹_{-2g} and E¹_{+2g} bands are found to be separated by ~ 5 cm⁻¹ both on the bottom, slope and top of the patterned MoS₂. This TERS measurement is repeatable as shown in the Fig. S18.

Unfortunately, we did not conduct TERS scan in the valley (the depth of 0 nm in Fig. 3d) in a diagonal direction. The TERS instrument to which we had access to was just newly installed and we had limited access to it. No additional experiments could be performed on our samples.

To emphasize that the strain arises from elongation rather than bending, we have added the following to the revised main text: “The tensile strain in the 2DM is induced through the elongation of the 2DM during the process of contact-transferring a planar 2DM onto silicon dioxide (SiO₂) dielectric surfaces ...”. As the elongation in

monolayer MoS₂ is relatively high in both the x and y lateral directions due to sinusoidal waves modulated in two directions, we do not observe local compressive strain typically associated with the bending of thin films.

Revisions:

- In Lines 80-89 of Introduction section, we added sentences to clarify the nature of the strain transfer as follows: “The tensile strain in the 2DM is induced through the elongation of the 2DM during the process of contact-transferring a planar 2DM flake using an elastomeric stamp onto a grayscale sinusoidal silicon dioxide (SiO₂) dielectric. This sinusoidal topography is previously fabricated by t-SPL and dry etching and can be programmed by adjusting the aspect ratios, also referred to as depth-to-pitch ratios. The depth-to-pitch ratio control capability of t-SPL, which cannot be so precisely achieved through other grayscale nanopatterning techniques such as electron beam lithography⁴⁸ and interference lithography,⁴⁹ provides deterministic control of strain induced in 2DMs. Varying the depth-to-pitch ratios of nanopatterns allows for the introduction of areas with strain gradients within a single 2DM flake by adjusting the amplitude and spatial frequency of the sinusoidal waves.”.
- The TERS data on the bottom of the sinusoidal structure was added in Fig. 3f.

Fig. 3| Strain characterization of strained monolayer MoS₂ transistors. ... d, AFM topography showing the strained transistor composed of sinusoidally patterned structures supporting the monolayer MoS₂ sheet as channel and the edge-contacted metal electrodes as Source and Drain. The region of TERS mapping is highlighted with a green rectangle. **e**, Topography image measured during hyperspectral TERS mapping with a step size of 50 nm. The position of the sinusoidal wave structure **on the bottom, the top and the slope** is marked with red squares and illustrated in the scheme below where the resolutions of micro-Raman spectroscopy and TERS are compared. **f**, Comparison of the average TERS spectra of the flat region and the strained regions **on the bottom, the top and the slope**. A splitting of the E_{12g}¹ peaks is observed in the strained region, confirming the presence of strain in the MoS₂ sheet.

(4) If the authors redo their analysis and still see >2x improvement from strain, then the authors should discuss why they observe more enhancement than other works.

a. What else is newly contributed from this work? Again, mobility enhancement from reduced intervalley scattering in strained 1L-MoS₂ is already documented.

Response: We appreciate the opportunity to clarify our approach and provide a more thorough explanation to underline the original aspects of our work.

As detailed above in the Comment 1 (page 1 of this letter), the observed 8x improvement in our work lies in the middle of previously reported values.

Firstly, in the Introduction section, we have included additional citations from state-of-the-art works to provide a more comprehensive overview of strain engineering of 2DMs. We have also cited the article mentioned in the comment. In the revised version of the manuscript, we have made it clearer that several new techniques have been

successfully developed or improved to induce strain and study its physical effects. Among these techniques, strain engineering of atomically thin materials on pre-patterned substrates has raised attention for applications of nanoscale-sized devices.

Additionally, we also further explained the advantages of using sinusoidal wave patterns. Compared to other approaches such as nanopillar arrays, rippled or crested substrates, smooth grayscale nanotopographies offer improved conformal attachment of 2DMs by reducing wrinkling of these atomically thin materials. This results in mechanically stable and improved dielectric-semiconductor interfaces. In contrast to monolayer 2DMs strained non-uniformly in sharp crested patterns, where strain is very high at peaks and very low on flat parts, grayscale nanopatterns offer a more homogeneous distribution of strain, while keeping strain uniformly distributed within the pattern area. The writing using tSPL brings the additional advantage to design deterministically the locations where the strain should occur.

Revisions:

- In Lines 58-73 of the Introduction section, we added a paragraph to distinguish our fabrication method as follows: “The use of thermally actuated micromechanical devices,²⁴ substrate heating,²⁵ thermomechanical nanoindentation,²⁰ thin-film stress induced by electric fields,²⁶ bulging devices with pressurized components,^{21,27} tip indentation to stretch 2DMs via direct contact,²⁸ and the bending and/or stretching of flexible polymer substrates²⁹⁻³³ have been employed to induce strain and study its physical effects in 2DMs. However, most of these techniques are incompatible with existing silicon-based technologies with high-density integration capabilities. Strain induced by substrate lattice mismatch,³⁴ thermal expansion mismatch,³⁵ integration with thin film stressors^{27,36} or underlying thin film stress constrains the choice of substrate. For scalability purpose, strain engineering of atomically thin materials using pre-patterned substrates has been developed.^{22,37-43} For instance, crested substrates have been introduced to enhance the performance of optical and/or electrical devices based on 2DMs with induced strain effects. However, the use of pre-structured substrates often results in suspended 2DMs, leading to potentially unstable semiconductor-dielectric interfaces. Therefore, the fabrication of compact nanoscale 2DM transistors with deterministic strain distribution compatible with advanced device architecture has yet to be achieved.”.
- In Lines 413-415 of the Conclusions section, we added a sentence to clarify the novelty in nanofabrication as follows: “Precisely patterned surface topography at the single-digit nanometer scale enables deterministic changes in the tensile strain induced in 2D materials, thereby locally altering their electrical and optical properties while offering a seamless device integration option.”.

b. Authors may want to consider benchmarking their devices against the literature to solidify the novelty of their work for the readers.

Response:

Related to your Comment 1 (page 1 of this letter), we added a table benchmarking the performances of strained MoS₂ FETs from the literature.

Revisions:

Same as the Comment 1 of Reviewer 1.

Minor:

(5) Have the authors conducted CV measurements on their patterned dielectrics?

Response:

The expected capacitance of a single FET with an area of $1 \times 2.2 \mu\text{m}^2$ and an average SiO₂ thickness of 140 nm is approximately 5.6×10^{-16} F. Sub-femtofarad capacitance measurements require dedicated hardware that is not available in our laboratory. This is due to the presence of parasitic capacitances which are much larger and severely impact the accuracy on the extracted value of the capacitance under test. As an example, these capacitance values are approximately four orders of magnitude lower than those reported in literature (<https://www.nature.com/articles/s41565-019-0361-x>). Consequently, in our work, the capacitance is determined through simulations and analytical calculations. We have added these values and more details of the calculations to the main text and Supplementary Information (Fig. S30-31).

(6) The authors say “edge-contacted S/D electrodes” in line 190, which could be misleading. Don’t the authors have regular top contacts achieved using lift-off? Maybe the reviewer misunderstood.

Response:

Thank you for this relevant observation. The 2D flake uncovered by PMMA resist after the second EBL step was etched using XeF₂ as the process (Step 11) shown in Fig. S8. Our analysis shows that the 2D flakes under the electrodes are probably removed as shown in Raman data in Fig. S12. However, we acknowledge that the Raman data alone is insufficient to conclusively determine that we have purely edge-contacted S/D electrodes.

In a recent paper published from our group (A. Conde-Rubio, et al. Edge-contact MoS₂ transistors fabricated using thermal scanning probe lithography, *ACS Appl. Mater. Interfaces* **14**, 42328 (2022)), we used a very similar etching process and demonstrated there the edge-contact structure using TEM. However, in this work here, we have not performed the TEM measurement and thus we don’t have a high resolution image of the contact configuration of 2DM and electrodes both with sinusoidal patterns. Therefore, we have removed the statement of edge-contacted S/D electrodes to avoid misunderstanding.

Revisions:

- In Line 145, “Source (S) and Drain (D) electrodes with edge-contact configuration...” was corrected as “Source (S) and Drain (D) electrodes...”
- In Line 222, “...a strained 2D FET with the edge-contacted S/D electrodes” was corrected as “...a strained 2D FET with the S/D electrodes”.
- A figure reporting the Raman line mapping on the transistor is added in SI.

Supplementary Fig. S12: (a) Optical image of the strained and the unstrained FETs made from the same monolayer MoS₂ flake. (b) The corresponding Raman line mapping result of the selected area.

(7) Authors state they fabricated and measured 400 devices, but do not show the statistics for all of them.

Response:

We fabricated about 10 nominally identical devices with 6 channel lengths and 6 patterning depths, as well as around 40 extra devices with 0.9 % strain and 0.65% strain for the statistics of performance (Fig. 4I). In total, around 400 devices were measured.

Revisions:

In Fig. 4, the legend was added with a sentence “The error bars in panels e-j represent the maximum and minimum values with the representative points corresponding to the average values for 120 transistors in panels e, i, and j, and for 30 transistors in panels f-h.”.

(8) There are also MOCVD and exfoliated devices, are they mixed in Fig. 4 or is this exfoliated alone? Maybe it would be best to separate them for transparency.

Response:

The data in Fig. 4 are based on the MOCVD grown monolayer MoS₂ transistors alone. In the revised version, we clarified it.

Revisions:

In Fig. 4, the legend was corrected as “**Fig. 4| Electrical characteristics of the strained MOCVD grown monolayer MoS₂ transistors. ...**”.

(9) “Mobility” needs to be explicitly stated as field-effect mobility in the figures.

Response:

In the revised version, we stated the field-effect mobility instead of mobility in the figures.

Revisions:

The y-axis titles of Fig. 4h, Fig. 5c and Fig. 5f-g were corrected as “Field-effect mobility (cm²V⁻¹s⁻¹)”. The y-axis titles of Fig. 5a-b were corrected as “Field-effect mobility enhancement”. The x-axis title of Fig. 4l was corrected as “Field-effect mobility (cm²V⁻¹s⁻¹)”.

(10) Some labels inside the figures say “strained” only instead of “X% strained” (e.g., Fig 4b,c, Fig. 5d).

Response:

In the revised version, we specified the strain value in the Figs 4 and 5.

Revisions:

In Figs 4 and 5, the strain values are specified.

(11) Fig. 5d should say E-E_c on the y-axis. Also, check MOCVD/exfoliated representative colors or labels in Fig. 5a,b,c (might be switched?)

Response:

In the revised version, we changed it to E-E_c since the zero corresponds to the bottom of the conduction band. We switched the colors in Fig. 5a to keep consistent with Fig. 5b&c. We also changed the colors in Fig. 5f and 5g, with strained data in red and flat data in blue to be consistent with all other graphs.

Revisions: In Fig. 5, the colors were modified to keep all the data consistent.

(12) Gate leakage is not shown anywhere.

Response:

We thank the reviewer for the comments. In the revised version, we added the gate leakage, I_{GS} versus V_{GS} in Fig. 4b.

Revisions:

In Lines 254-255, we added a sentence “The gate leakage currents in all the measured transistors are in the order of a few pA, i.e., much smaller than the drain currents in the ‘on’ region.”

References

- [1] <https://doi.org/10.1021/acs.nanolett.2c01707>
- [2] <https://doi.org/10.1021/acsnano.3c10495>
- [3] <https://arxiv.org/abs/2309.10939>
- [4] <https://doi.org/10.1021/acsnano.3c03626>
- [5] <http://dx.doi.org/10.1088/0022-3727/48/37/375104>
- [6] <https://doi.org/10.1109/TED.2015.2461617>
- [7] <https://doi.org/10.1021/acs.nanolett.3c05091>
- [8] <https://doi.org/10.1021/acsnano.6b01149>
- [9] <https://doi.org/10.1063/1.4868536>
- [10] <https://doi.org/10.1002/sml.202100940>
- [11] <https://doi.org/10.1002/adma.201806020>
- [12] <https://doi.org/10.1038/ncomms10908>

Response: We analyzed all the references above and cited the most relevant ones [1-6] in the Supplementary Table S3. And detailed explanations and comparisons are provided in the response to the Comment 1 (page 1 of this letter).

Reviewer #2 (Remarks to the Author):

Liu and coworkers reported a new strain engineering approach to apply biaxial tensile strain to MoS₂ and investigated how it impacts the electrical transport properties of MoS₂ FETs made out of it. I found this work well-presented and the strain engineering technique they introduced interesting. I believe it is appealing to a board audience working on strain engineering of 2D materials and 2D materials-based FETs. However, the below comments and questions should be addressed before the manuscript can be considered for publication.

Response: We thank the reviewer for these positive comments.

1. It is not the first time strain has been applied to 2D materials by conforming 2D flakes onto patterned substrate. For example, Hinnefeld, J.H., Gill, S.T. & Mason, N. Graphene transport mediated by micropatterned substrates. *Appl. Phys. Lett.* 112, 173504 (2018). The authors should explain what is the advantage of using the sinusoidal wave pattern and why not using other periodic patterns other than sinusoidal? Besides the amplitude, does the wavelength of the pattern impact the strain transfer and the mobility?

Response:

We thank the reviewer for the comments. We have updated our introduction with detailed state-of-the-art literature review and discuss advantages of using sinusoidal waves.

The strain can be introduced into 2D materials by conforming 2D flakes onto patterned substrates, such as with ripple array (*Nature Electronics* **5**, 489–496 (2022)) or nanocone array (*Nature Communications* **6**, 7381 (2015)). However, the strain value in these two previous works is distributed randomly without uniform value. In our work, we demonstrate that a sinusoidal wave pattern with a smooth surface allows the strain value to be uniformly distributed in the 2D channel. In the revised version, we added a paragraph to do a thorough review on strain engineering of 2D materials. Key benefits of sinusoidal patterning of gate oxide are summarized below:

- We developed an approach for fabricating **permanently strained** 2DM transistors at the nanoscale using a **deterministic gray-tone topography** patterning of gate dielectric for device integration purposes. This deterministic grayscale nanopatterning is achieved by thermal scanning probe lithography (t-SPL) with single-digit nm resolution, which cannot be achieved through other grayscale nanopatterning techniques such as E-beam lithography and interference lithography.
- The surface of the gate dielectric patterned with sinusoidal wave topographies **induces localized tensile strain** in the monolayer MoS₂ flake transferred on top.
- The precise depth-to-pitch ratio control of nanotopographies provides **deterministic control of strain induced in 2DMs**.
- The newly developed nanopatterning allows introducing **strain gradients within the same 2DM flake** by adjusting the amplitude and spatial frequency of the sinusoidal waves in a controllable way.
- Grayscale nanotopographies provides improved **conformal attachment of 2DMs** by reducing wrinkling and free-standing of these atomically thin materials. This results in a **mechanically stable** environment for further fabrication processes (e.g., resist coating for 2DM patterning, dielectric deposition, and lift-off metallization) and **improved dielectric-semiconductor interfaces**.
- Unlike sharp crested patterns, grayscale nanopatterns results in a more **homogeneous distribution of strain** while keeping the strain **localized within the pattern area**.

The strain value depends on the depth-to-wavelength ratio. Smaller wavelength allows to achieve higher strain in 2D materials. In a recent paper published from our group (Erbaş et al. *Microsystems & Nanoengineering* **10**, 28 (2024)), we showed that induced strain on monolayer 2DM depends on depth-to-pitch ratio (amplitude-to-wavelength ratio). In this work, we used a constant pitch of 300 nm and different depths to obtain the different strain values.

Revisions:

- In Lines 58-73 of the Introduction section, we added a paragraph to distinguish our fabrication method as follows: “**The use of thermally actuated micromechanical devices,²⁴ substrate heating,²⁵ thermomechanical nanoindentation,²⁰ thin-film stress induced by electric fields,²⁶ bulging devices with pressurized**

components,^{21,27} tip indentation to stretch 2DMs via direct contact,²⁸ and the bending and/or stretching of flexible polymer substrates²⁹⁻³³ have been employed to induce strain and study its physical effects in 2DMs. However, most of these techniques are incompatible with existing silicon-based technologies with high-density integration capabilities. Strain induced by substrate lattice mismatch,³⁴ thermal expansion mismatch,³⁵ integration with thin film stressors^{27,36} or underlying thin film stress constrains the choice of substrate. For scalability purpose, strain engineering of atomically thin materials using pre-patterned substrates has been developed.^{22,37-43} For instance, crested substrates have been introduced to enhance the performance of optical and/or electrical devices based on 2DMs with induced strain effects. However, the use of pre-structured substrates often results in suspended 2DMs, leading to potentially unstable semiconductor-dielectric interfaces. Therefore, the fabrication of compact nanoscale 2DM transistors with deterministic strain distribution compatible with advanced device architecture has yet to be achieved.”.

- In Lines 78-96 of the Introduction section, we added sentences to clarify the novelty of our work as follows: “The t-SPL with a lateral resolution below 10 nm with sub-nanometer vertical depth control enables fabrication of high-resolution grayscale nanopattern with deterministic aspect ratio control.^{45,47} The tensile strain in the 2DM is induced through the elongation of the 2DM during the process of contact-transferring a planar 2DM flake using an elastomeric stamp onto a grayscale sinusoidal silicon dioxide (SiO₂) dielectric. This sinusoidal topography is previously fabricated by t-SPL and dry etching and can be programmed by adjusting the aspect ratios, also referred to as depth-to-pitch ratios. The depth-to-pitch ratio control capability of t-SPL, which cannot be so precisely achieved through other grayscale nanopatterning techniques such as electron beam lithography⁴⁸ and interference lithography,⁴⁹ provides deterministic control of strain induced in 2DMs. Varying the depth-to-pitch ratios of nanotopographies allows for the introduction of areas with strain gradients within a single 2DM flake by adjusting the amplitude and spatial frequency of the sinusoidal waves. Compared to other approaches such as nanopillar arrays and rippled or crested substrates, grayscale nanotopographies offer also improved conformal attachment of 2DMs by reducing wrinkles and suspended parts. This results in improved dielectric-semiconductor interfaces and a mechanically more stable environment compatible with subsequent fabrication processes. In contrast to the 2DM strained by sharp crested patterns, where strain is non-uniform and is very high at peaks and very low on flat parts, grayscale nanopatterns offer a more homogeneous distribution of strain while still keeping the strain localized within the pattern area.”.
- In Lines 413-415 of the Conclusions section, we added a sentence to clarify the novelty in nanofabrication as follows: “Precisely patterned surface topography at the single-digit nanometer scale enables deterministic changes in the tensile strain induced in 2D materials, thereby locally altering their electrical and optical properties while offering a seamless device integration option.”.

2.The strained devices and the unstrained devices in this work have different gate and channel structure (sinusoidal patterned gate vs flat gate; sinusoidal-shaped channel vs straight channel). Both the gate and the channel structure may significantly impact the electrical behavior of the device. The authors briefly addressed this by claiming the difference of the gate capacitance is small. The authors should give a more detailed explanation of how they calculated the gate capacitance and how they define the channel length. Also, the claim that 'the capacitance of the strained FET is also same as that of the flat one' in Supplementary Section 7 is not valid and contradicts the simulation result in Fig. S24.

Response:

As shown in Fig. 4a, the strained FET device consists of sinusoidal patterned dielectric and thus sinusoidal-shaped channel, while the gate electrode is the flat silicon substrate for both the strained FETs and the unstrained FETs. Hence, the gate electrodes of the strained and flat FETs are both flat.

We discussed the calculation of gate capacitance in details in Section 7 of Supplementary Information. The analytical calculation of the gate capacitance at specific dielectric heights of the strained transistor is calculated using the equation: $C_{ox} = \epsilon_0 \epsilon \Sigma [(E(y))^2 \Delta y] / V_{GS}^2$; where V_{GS} is the voltage of back gate, ϵ is the dielectric constant, $E(y)$ is the electrical field at the position y . The difference of the gate capacitance is small since our comparison is based on flat dielectrics with a 140 nm gate dielectric thickness (for the 0.9% strained FET) and sinusoidal dielectrics with a mean thickness of 140 nm (and a depth up to 60 nm).

The channel length is calculated based on the elongation of 2D flakes on the sinusoidal patterned substrate. Since the highest strain is only 1%, the variation of channel length is negligible.

We have modified the main text to avoid misunderstanding on the difference of capacitance between strained and flat FETs.

Revisions:

- In Lines 294-303, we added a paragraph to clarify the difference in gate capacitance as follows:

“Analytical calculations and simulations of the electrical field distribution and capacitance in the strained transistors show no significant difference in the gate capacitance, indicating that it exerts no discernible impact on the strained transistors. While flat transistors exhibit a gate capacitance of 2.47×10^{-8} F/cm², that of strained transistors with a 60 nm peak-to-peak depth is 2.56×10^{-8} F/cm² (Supplementary Fig. S30 and Fig. S31). The variation in gate dielectric thickness leads to differing doping concentrations in the FET channel, impacting mobilities. However, our comparison is based on flat dielectrics with a 140 nm gate dielectric thickness and sinusoidal dielectrics with a mean thickness of 140 nm, both having the same surface quality. This results in a doping concentration change of only 3.5%, which does not introduce significant alterations, according to theoretical and experimental studies.⁵⁶”.

- In the revised SI, we modified the section 7 of Electrical field distribution in strained FETs as follows:

“The analytical calculation of the capacitance at specific dielectric heights of the strained transistor is calculated using the equation:

$$C_{ox} = \frac{1}{V_{GS}^2} \epsilon_0 \epsilon \sum [E(y)^2 \Delta y] \quad (S9)$$

where V_{GS} is the voltage of the back gate, ϵ is the dielectric constant, $E(y)$ is the electrical field at the position y . Along the longitudinal direction, extending from the gate electrode to the FET channel, we initially simulated the electric field distribution using COMSOL Multiphysics (version 6.0) with electrostatics physics modeling. Subsequently, we extracted the electric field norm distribution at specific positions of the sinusoidal design modulated in two dimensions $f(x,z)$, such as dielectrics with heights of 110 nm, 140 nm, and 170 nm, and performed capacitance calculations in MATLAB (version R2020b) utilizing the aforementioned formula. Furthermore, we compared our capacitance calculations obtained through combined simulation and analytical approaches with parallel plate capacitance assumptions at heights of 110 nm, 140 nm, and 170 nm, and observed deviations in the results of less than 3%. Local capacitances in the sinusoidal design exhibited a proportional relationship to the parallel plate approach, with the mean values of calculated gate capacitance across different heights equating to the capacitance at 140 nm.

Finally, when the Maxwell capacitance of our grayscale design in COMSOL Multiphysics simulation was computed, it was only 3.5% higher than that of the flat design, as shown in Figure S31.”

3. It is well-known that the strain may adjust the bandgap of MoS₂. Could the authors comment on what is the impact of the bandgap of MoS₂ in their strained devices?

Response:

We have considered it in the revised version by adding Supplementary Fig. S19 for the experimental results of bandgap modulation obtained through photoluminescence (PL) measurements and its corresponding micro-Raman spectroscopy characterization on sinusoidal nanopatterns. We observe a ~ 38 meV shift for a 1.3cm^{-1} shift in the E^1_{2g} peak position, corresponding to ~ 132 meV/% strain and we have updated Fig 5e caption where we predict ~ 150 meV/% strain shown by our first-principles simulations.

Revisions:

- In Lines 359-360, we added a sentence “The energy separation variation rate is around 150 meV/% strain, close to the experimental obtained value from the photoluminescence measurement (Supplementary Fig. S19).”.
- In Lines 399-400 of the legend of Fig. 5e, we added a sentence “See Supplementary Fig. S19 for the experimental results on the strain induced bandgap modulation obtained through photoluminescence (PL) measurements.”.
- We added the Supplementary Fig. S19 in SI to analyze the shift rate of A-exciton.

Supplementary Fig. S19: (a) Optical image of the exfoliated MoS₂ flake transferred onto a nanoengineered substrate. (b) Micro-Raman spectroscopy characterization of strained 1L MoS₂ on sinusoidal nanopattern modulated in two dimensions and its (c) photoluminescence (PL) characterization to visualize bandgap modulation of MoS₂ monolayer under biaxial tensile strain. We observe a ~38 meV shift for a 1.3 cm⁻¹ shift in the E_{2g}¹ peak position, corresponding to ~132 meV/% strain.

4. In the literature review section for recent strain engineering techniques of 2D materials, some popular strain engineering methods are omitted. For example, using thin film stressors (A. Azizimanesh, T. Peña, A. Sewaket, W. Hou, and S. M. Wu, *Appl. Phys. Lett.* 118, 213104 (2021); W. Hou, A. Azizimanesh, A. Dey, Y. Yang, W. Wang, C. Shao, H. Wu, H. Askari, S. Singh, S. M. Wu, *Nature Electronics* 7, 8 (2024)). The authors may consider having a more thorough literature review.

Response:

In the revised version, we have added a paragraph for a more detailed literature review of strain engineering of 2D materials. We benchmarked the techniques from the most impactful articles, including the two articles suggested by the reviewer.

Revisions:

- We added a paragraph to distinguish our fabrication method as we answered to your comment 1.
- We have updated our introduction to provide a more detailed literature review as we answered to your comment 1.
- The papers that you recommended are cited as References 27 and 36.

Ref 27: W. Hou, A. Azizimanesh, A. Dey, Y. Yang, W. Wang, C. Shao, H. Wu, H. Askari, S. Singh, S. M. Wu, *Nature Electronics* 7, 8 (2024))

Ref 36: A. Azizimanesh, T. Peña, A. Sewaket, W. Hou, and S. M. Wu, *Appl. Phys. Lett.* 118, 213104 (2021)

Reviewer #3 (Remarks to the Author):

In this manuscript, the authors developed a sophisticated nanofabrication for mobility enhancement in 2DM transistors, with the biaxial tensile strain induced by contact transfer of the 2DM layer on a grayscale nanopatterned SiO₂ dielectric. The SiO₂ pattern was fabricated by a novel technique of t-SPL and dry etching, which can be integrated into the existing silicon-based technology.

Under the tensile strain, the monolayer MoS₂ FETs exhibited a noticeable increase of electron mobility, exceeding 180 cm²/Vs at room temperature and improved by over a factor of 8 while comparing the unstrained (planar) devices. Additionally, the first-principle calculations are conducted, to analyze and validate the mobility improvement, with respect to the intervalley scattering and coupling.

The nanotechnology and fabrication are intriguing, and demonstrates their superior mobility and electrical characteristics.

Based on these points, I am positive in its possible publication in nature communications, after sufficiently addressing the following questions:

Response: We thank the reviewer for these positive comments.

Comment 1. Strain engineering techniques in MoS₂ and 2DM shall be shortly stated in the introduction.

Response:

We have updated our introduction to provide a more detailed literature review of strain engineering of 2D materials, as other reviewers also suggest. In the revised version, we have mentioned various techniques and discussed the advantages and limitations of different techniques.

Revisions:

In Lines 58-73 of the Introduction section, we added a paragraph to provide a more comprehensive overview of strain engineering in 2DMs as follows: “The use of thermally actuated micromechanical devices,²⁴ substrate heating,²⁵ thermomechanical nanoindentation,²⁰ thin-film stress induced by electric fields,²⁶ bulging devices with pressurized components,^{21,27} tip indentation to stretch 2DMs via direct contact,²⁸ and the bending and/or stretching of flexible polymer substrates²⁹⁻³³ have been employed to induce strain and study its physical effects in 2DMs. However, most of these techniques are incompatible with existing silicon-based technologies with high-density integration capabilities. Strain induced by substrate lattice mismatch,³⁴ thermal expansion mismatch,³⁵ integration with thin film stressors^{27,36} or underlying thin film stress constrains the choice of substrate. For scalability purpose, strain engineering of atomically thin materials using pre-patterned substrates has been developed.^{22,37-43} For instance, crested substrates have been introduced to enhance the performance of optical and/or electrical devices based on 2DMs with induced strain effects. However, the use of pre-structured substrates often results in suspended 2DMs, leading to potentially unstable semiconductor-dielectric interfaces. Therefore, the fabrication of compact nanoscale 2DM transistors with deterministic strain distribution compatible with advanced device architecture has yet to be achieved.”.

Comment 2. Although the AFM image shows the adhesion map and presence of the strained 2DM on the dielectric surface (Fig. 2i), the transfer quality with respect to the sheet size, defect, crack or wrinkle etc. needs to be included.

Response:

In the revised version, we added more discussions about the presence of a few visible wrinkles in Fig. 2h-j and the transfer quality in the main text. Regarding the sheet size, Fig. 2d shows a typical MOCVD grown triangular flake with the side length of 80 μm. We used the MOCVD grown flakes with side lengths of 50-100 μm as shown in Supplementary Fig. S5(a). The transfer quality with respect to defects was characterized by SEM and Raman in Fig. S5(b) and S5(c). The transfer quality with respect to cracks or wrinkles was characterized in Fig. 2h-j. More data are provided in the SI. We exfoliated monolayer flakes as shown in Supplementary Fig. S10(a).

Revisions:

- In Line 160, a sentence was added as “However, a few wrinkles with tens of nanometers width are also shown in Fig. 2j.”.

- In Lines 180-181, the legend “AFM deformation image indicating conformal contact between the 2D flake and SiO₂ substrate” was corrected as “AFM deformation image indicating conformal contact between the 2D flake and SiO₂ substrate with exception of a few wrinkles with ≤ 6 nm deformations”.
- Regarding the improvement of the flake quality, we added a Supplementary Fig. S6 to compare the quality of the flake before and after the improvement in the transfer technique.

Supplementary Fig. S6: The SEM images show (a) a lot of cracks and (b) no cracks before and after the improvement in the technique of 2D flake transfer, respectively. The improvement was made in steps d-e in Fig. S4.

Comment 3. With a 65 nm peak-to-peak amplitude of the SiO₂ pattern, a 1% strain which corresponds to a shift of the E_{12g} peak by -4.4 cm⁻¹, was measured and extracted, rather than a 2.8% strain as theoretically expected. This large discrepancy (the 2-3 times lower than the maximum achievable limit) shall be clarified by the authors, regarding the SiO₂ pattern and the 2DM layer map.

Whether the 2DM layer is domestically, effectively adhered to surface of the patterned SiO₂ dielectric? Can the sinusoidal topography of 2DM layer be provided ?

Response:

In the revised version, we clarified this issue by adding a paragraph. The large discrepancy between the calculated strain and measured strain derives from several factors, including the flake sliding and strain relaxation during transfer, as well as imperfect attachment of the 2DM on a sinusoidal surface with a high spatial frequency pitch.

In our fabricated sample, we measured a shift of the E_{12g} peak by -4.4 cm⁻¹, corresponding to a strain of 1% based on the widely reported and theoretically predicted Raman peak shift of -4.5 cm⁻¹/% strain for E_{12g} phonon. However, other works have reported peak redshift values ranging from 2.1 cm⁻¹/% strain¹ to 5.2 cm⁻¹/% strain for E_{12g} phonons. These minimum and maximum values of our work lead to an estimated strain ranging from 0.85% to up to 2.10%. While it is possible to derive intermediate values from existing literature, we opted to utilize the extensively documented and theoretically anticipated Raman peak redshift of 4.5 cm⁻¹/% strain for the E_{12g} phonon as our benchmark.

The SEM, AFM and TEM images in Fig. 2 indicate that the 2DM layer conformally adheres to the surface of the patterned SiO₂ dielectric. The SEM (Fig. 2e and Fig. S13) and TEM images (Fig. 2e and Fig. S11) qualitatively show the 2DM layer adheres to the sinusoidal patterned substrate. The element mappings of the cross-section transistors in Fig. 2g show continuous wavy monolayer MoS₂ layer that has intimate contact with the patterned SiO₂ substrate. Supplementary Fig. S11 shows a cross-sectional TEM image of the strained transistor’s channel showing the monolayer MoS₂ flake follows the wavy SiO₂ substrate with intimate contact.

AFM images of the monolayer MoS₂ flake on the patterned substrates were taken in PeakForce QNM® mode using the Multimode (Bruker) Scanhead and Nanoscope V controller (Bruker). ScanAsyst-Air cantilevers with a spring constant of 0.4 N/m were utilized, and peak forces were set to 10 nN for quantitative mechanical characterization. Notably, the presence of 2DM is scarcely discernible in both the topography and deformation maps, indicating a conformal adhesion between the 2DM and the dielectric surface (Fig. 2h and 2j). In the revised version of the manuscript, we emphasized the significance of using relatively high tip forces of 10 nN for monolayer flakes to obtain deformation maps through quantitative nanomechanical AFM characterization.

Therefore, AFM topography and adhesion images, TEM images, and EDX maps have provided sinusoidal topography of the 2D material layer on patterned SiO₂ to support adhesion and topographies of 2D layers.

Additionally, we also adapted a few specific steps in the device fabrication to protect the strained 2DM layer during the fabricating process, the 2DM layer is always covered by a PMMA resist layer until the final lift-off step (Supplementary Fig. S8-10).

Revisions:

- In Lines 206-217, we added a paragraph to clarify the discrepancy in the calculated strain and measured strain as follows “While it is possible to derive intermediate values from existing literature,^{20,22,30} we opted to utilize the extensively documented and theoretically anticipated Raman peak redshift of 4.5 cm⁻¹/% strain for the E_{12g} phonon as our benchmark. The discrepancy between the theoretically calculated strain, which is related to surface area increase through sinusoidal nanopatterning, and measured strain might arise from several factors, including flake sliding and strain relaxation during transfer, as well as imperfect attachment of the 2DM on a sinusoidal surface with a high spatial frequency pitch. During the elongation of the 2D flake, the sliding between the 2D layer and the sinusoidal patterned substrate is inevitable due to the weak van-del-Waals force,⁵¹ which is one of the critical challenges in strain engineering of 2DMs. The transfer of 2D flakes involves a few temperature-related steps (see Supplementary Fig. S4 and S7) that cause strain relaxation of the 2D flakes. The 1% strain achieved remains relatively high, which is sufficient to approach the upper mobility limits in MoS₂ according to first-principles calculations.”
- In Lines 158-159, “Notably, the presence of 2DM is scarcely discernible in both the topography and deformation maps, ...” was corrected as “Notably, the presence of 2DM is scarcely discernible in both the topography and deformation maps, which are characterized by relatively high tip forces of 10 nN for monolayer flakes, ...”
- In Lines 494-498 of the Methods section, we added the parameters of AFM measurement as follows: “AFM images of the monolayer MoS₂ flake on the grayscale substrates were taken in PeakForce QNM® mode using the Multimode (Bruker) Scanhead and Nanoscope V controller (Bruker). ScanAsyst-Air cantilevers with a spring constant of 0.4 N/m were utilized, and peak forces were set to 10 nN for quantitative mechanical characterization.”

Comment 4. Hysteresis-free transfer characteristics (Fig. 4d) have been observed in the strained MoS₂ transistor, low-frequency noise characterization can be included to identify the semiconducting layer and interface quality, as well.

Response:

At the time when we took the data, we were not aware of potential relevance of low-frequency noise measurement. We found a few papers that have reported this type of measurements, such as (X. Xie, et al. Low-frequency noise in bilayer MoS₂ transistor, *ACS Nano* **8**, 5633 (2014)), (Q. Gao, et al. Improved low-frequency noise in CVD bilayer MoS₂ field-effect transistors, *Appl. Phys. Lett.* **118**, 153103 (2021)), but most papers focusing on mobility of MoS₂ transistors do not report such measurement, such as the cited references [1,4-5,11-12,17,40-43,56].

However, we provided all the related details of AFM nanomechanical characterization. In addition to AFM topography, TEM images, and EDX maps to characterize the interface quality of the 2D semiconducting layer.

Comment 5. The authors state that the electron mobility enhancement in the monolayer MoS₂ transistor, from 60 cm²/Vs measured in the unstrained to 150 cm²/Vs in the 0.65%-strain and 180 cm²/Vs in the 0.90%-strain, because of the drastic reduction in intervalley scattering due to the induced strain.

However, as theoretically predicted, the mobility of the single-layer MoS₂ is calculated to be reached up ~410 cm²/Vs, where the intrinsic phonon scattering is dominated (10.1103/PhysRevB.85.115317). As well, the experimental value of ~200 cm²/Vs has been reported for the single-layer MoS₂, while considering additional scattering mechanisms, e.g. the non-intrinsic Coulomb scattering.

In this manuscript, considering of the t-SPL and etching process to fabricate the gray-tone topography, the surface topography of the biaxial sinusoidal pattern and their impact on intrinsic and non-intrinsic scattering and mobility, shall be discussed and clarified, from the aspects of experiments and simulations.

Response:

The field-effect mobility of the single-layer MoS₂ transistor is calculated to up 410 cm²/Vs, however, the experimental mobility is much smaller due to non-intrinsic Coulomb scattering. The high experimental value of ~200 cm²/Vs has been reported in the paper (B. Radisavljevic et al., Single-layer MoS₂ transistors, *Nature Nanotechnology* **6**, 147–150 (2011)) thanks to the high dielectric constant of the hafnium oxide gate dielectric and surface encapsulation. However, except for the Nat. Nanotechnol paper, such high field-effect mobility has been reported very rarely. In the revised version, we benchmarked the key parameters that are relevant to assess the performance of strained and unstrained monolayer MoS₂ FETs, including field-effect mobility, in Supplementary Table S3.

The non-intrinsic scattering is mainly caused by impurities within the MoS₂ layer itself and at the relatively rough underlying gate dielectric interface. The tensile strain induced in the semiconducting channel by underlying grayscale nanotopographies, as presented in this work, results in a remarkable increase in mobility even on SiO₂ dielectric, even with a global back gate, which could be advantageous in sensors with non-protected top surfaces.

In our first-principles simulations, we explored the impact of applying uniform tensile strain across the material, bypassing the direct simulation of intricate topographical features due to their computational complexity at the atomistic level. These features demand prohibitively large computational resources, making it impractical to replicate realistic patterns. Our focus lies on analyzing the effects of this uniform strain, particularly its influence on intrinsic scattering mechanisms and electron mobility. We demonstrate that strain plays a pivotal role, primarily by inducing a shift in the relative positions of the K and Q valleys, alongside investigating its effects on phonons and electron-phonon interactions. This shift gradually reduces and ultimately suppresses electron-phonon intervalley scattering, which significantly hampers electron mobility.

Regarding extrinsic contributions, particularly charged impurities at lower temperatures, we incorporate them through a charged impurity model. This model remains independent of strain effects but facilitates the inclusion of an average realistic contribution from such defects, accommodating temperature effects, and ultimately allowing a reconciliation between theoretical and experimental mobilities presented in this work.

While defects and extrinsic scattering factors are expected to minimally impact mobility in flat systems due to the primary role of valley distances and intervalley scattering, under strain, as valleys shift, the influence of extrinsic contributions becomes more significant and challenging to accurately model. Notably, the study of strain effects on impurity scattering remains intertwined with unknown experimental factors (e.g., is the impurity concentration different from one sample to the other or for different strains), as explicitly acknowledged in the manuscript.

In summary, our focus is on understanding the impact of tensile strain on mobility, acknowledging that it does not directly replicate the intricate surface features. Nonetheless, the essential features are captured, enabling us to interpret experimental results and provide a microscopic explanation for observed mobility enhancements. While we extensively investigate the intrinsic contributions of strain, the full exploration of extrinsic contributions, especially considering the complex interplay with realistic surface topography and substrate effects, lies beyond the scope of this study due to computational constraints at the atomistic level.

Revisions:

- In Lines 386-388, we added “Importantly, defects **within the MoS₂ layer itself and at the underlying gate dielectric interface, which cause extrinsic scattering**, are also predicted to play a role in the variations of mobility under strain.”
- In Lines 392-393, we also updated Fig. 5 and its caption “Mobility enhancement as a function of tensile strain for the strained exfoliated flake and strained MOCVD grown flake, comparing with the **DFT-calculated value at doping concentrations of 5×10^{12} cm⁻², excluding impurity effects to illustrate the enhancement limit for phonon-limited mobility.**”
- We also changed the colors in Fig. 5f and 5g, with strained data in red and flat data in blue to be consistent with all other graphs. In Fig. 5, the colors were modified to keep all the data consistent.

- We added Supplementary Table S3 showing comparison of performance of strained and unstrained single-layer MoS₂ FETs.

Supplementary Table S3: Comparison of performance of strained MoS₂ FETs

Refs	Material Thickness Method	Strain type	Strain value [%]	Raman shift [cm ⁻¹]	Carrier density [cm ⁻²]	Strained FET		Unstrained FET [cm ² V ⁻¹ s ⁻¹]
						Mobility [cm ² V ⁻¹ s ⁻¹]	Enhancement	
Experimental results								
[1]	MoS ₂ 1L CVD grown	Uniaxial tensile strain	0 ~ 0.7	1.61	1.1 × 10 ¹³	14 for 1L MoS ₂ at 0.7% strain	2x	7
[2]	MoS ₂ 1L CVD grown	Uniform biaxial tensile strain	0.7	E _{2g} ¹ : 3.3	NA	15.94	1.8x	8.71
[3]	MoS ₂ 1L CVD grown	Biaxial tensile strain	0.23	NA	NA	9.1 for 1L MoS ₂ at 0.23% strain	1.7x	5.4
[4]	MoS ₂ 1L and 2L Exfoliated	Uniaxial tensile strain	0.73-1.7	E _{2g} ¹ : 3.36-7.82	2 × 10 ¹²	127 for 1L MoS ₂ at 0.87% strain 82 for 2L MoS ₂ at 1.36% strain	2.52x 1.64x	50.5

[R1]	MoS ₂ 1L and 3L Exfoliated	Biaxial tensile strain	NA	0	9×10 ¹²	448 for 1L MoS ₂ 900 for 3L MoS ₂	20x (1L) 100x (3L)	20 (1L) 9 (3L)
[R2]	MoS ₂ 1L CVD grown	Local biaxial tensile strain	0.2-1.3	E _{2g} ¹ : ~6.5 A _{1g} : ~3.5	(0.8-1.8) ×10 ¹²	32	60x	0.5
[R3]	MoS ₂ 1L Exfoliated	Tensile strain	~0.7	NA	7.8×10 ¹²	~1150 for 1L MoS ₂ at 0.7% strain	100x	12.5
Our work	MoS ₂ 1L Exfoliated and CVD grown	Multiaxial tensile strain	0 ~ 1	4.4	3.7×10 ¹²	185 for 1L MoS ₂ at 0.9% strain	8x	23
Theoretical calculation results								
[5]	MoS ₂ 1L	Biaxial tensile strain	5	NA	1×10 ¹²	45	1.53x	30
[6]	MoS ₂ 1L	Biaxial tensile strain	0.4	NA	1×10 ¹²	NA	1.05x	NA
[R4]	MoS ₂ 1L	Uniaxial strain	2	NA	NA	0.9	6x	0.15
Our work	MoS ₂ 1L	Uniform tensile strain	0 ~ 1	NA	5×10 ¹²	485 for 1L MoS ₂ at 1% strain	8.46x	57.3

[1] Isha M. Datye, Alwin Daus, Ryan W. Grady, Kevin Brenner, Sam Vaziri, and Eric Pop. Strain-Enhanced Mobility of Monolayer MoS₂. *Nano Lett.* **22**, 8052–8059 (2022)

[2] Heechang Shin, Ajit Kumar Katiyar, Anh Tuan Hoang, Seok Min Yun, Beom Jin Kim, Gwanjin Lee, Youngjae Kim, JaeDong Lee, Hyunmin Kim, and Jong-Hyun Ahn. Nonconventional Strain Engineering for Uniform Biaxial Tensile Strain in MoS₂ Thin Film Transistors. *ACS Nano* **18**, 4414–4423 (2024)

[3] Jerry A. Yang, Robert K. A. Bennett, Lauren Hoang, Zhepeng Zhang, Kamila J. Thompson, Antonios Michail, John Parthenios, Konstantinos Papagelis, Andrew J. Mannix, Eric Pop. Biaxial Tensile Strain Enhances Electron Mobility of Monolayer Transition Metal Dichalcogenides. <https://doi.org/10.48550/arXiv.2309.10939>

[4] Yang Chen, Donglin Lu, Lingan Kong, Quanyang Tao, Likuan Ma, Liting Liu, Zheyi Lu, Zhiwei Li, Ruixia Wu, Xidong Duan, Lei Liao, and Yuan Liu. Mobility Enhancement of Strained MoS₂ Transistor on Flat Substrate. *ACS Nano* **17**, 14954–14962 (2023)

[R1] Hong Kuan Ng, Du Xiang, Ady Suwardi, Guangwei Hu, Ke Yang, Yunshan Zhao, Tao Liu, Zhonghan Cao, Huajun Liu, Shisheng Li, Jing Cao, Qiang Zhu, Zhaogang Dong, Chee Kiang Ivan Tan, Dongzhi Chi, Cheng-Wei Qiu, Kedar Hippalgaonkar, Goki Eda, Ming Yang, and Jing Wu. Improving Carrier Mobility in Two-Dimensional Semiconductors with Rippled Materials. *Nat. Electron* **5**, 489–496 (2022)

[R2] Arijit Kayal, Sraboni Dey, Harikrishnan G., Renjith Nadarajan, Shashwata Chattopadhyay, and Joy Mitra. Mobility Enhancement in CVD-Grown Monolayer MoS₂ Via Patterned Substrate-Induced Nonuniform Straining. *Nano Lett.* **23**, 6629–6636 (2023)

[R3] Tao Liu, Song Liu, Kun-Hua Tu, Hennrik Schmidt, Leiqiang Chu, Du Xiang, Jens Martin, Goki Eda, Caroline A. Ross, and Slaven Garaj. Crested Two-Dimensional Transistors. *Nat. Nanotechnol.* **14**, 223–226 (2019)

[5] Manouchehr Hosseini, Mohammad Elahi, Mahdi Pourfath, and David Esseni. Strain Induced Mobility Modulation in Single-Layer MoS₂. *J. Phys. D: Appl. Phys.* **48**, 375104 (2015)

[6] Manouchehr Hosseini, Mohammad Elahi, Mahdi Pourfath. Strain-Induced Modulation of Electron Mobility in Single-Layer Transition Metal Dichalcogenides MX₂ (M = Mo, W; X = S, Se). *IEEE Transactions on Electron Devices* **62**, 3192 – 3198 (2015)

[R4] Thibault Sohier, Marco Gibertini, Davide Campi, Giovanni Pizzi, and Nicola Marzari. Valley-Engineering Mobilities in Two-Dimensional Materials. *Nano Lett.* **19**, 3723–3729 (2019)

Authors' response to the second round of comments by referee #3

Reviewer #3 (Remarks to the Author):

Additional concern, for the strained and flat FETs, the contact resistances extracted by TLM (Fig. S26) contradict the results of YFM (Fig. S27). Any explanation? the statistics shall be discussed.

Thanks for the comments. Yes, we see a difference in contact resistances (R_c) calculated using the two methods: transfer length method (TLM) and Y-function method (YFM). The two values are still in the same order. The difference derives mainly from two aspects. (a) R_c calculated by TLM is extracted by linearly plotting total resistance versus channel length of six devices with various channel length (Fig. S26). However, the R_c calculated by YFM only focuses one device. (b) In the equation S7, the intrinsic mobility attenuation factor, θ_0 is considered negligible. Therefore, the TLM is more widely used and accurate to calculate R_c . The statistics of R_c has been discussed in Fig. 4g in the submitted revised version.